mathematical finance/mathematical physics

Bitcoin, k-means clustering, multiscale bubble indicator, log-periodic power-law singularity analysis, forecasting, market crashes

**Author for correspondence:**
J. C. Gerlach
e-mail: jgerlach@ethz.ch

# Dissection of Bitcoin's multiscale bubble history from January 2012 to February 2018

## J. C. Gerlach[1], G. Demos[1] and D. Sornette[1,2]

[1]Department of Management, Technology and Economics ETH Zürich, Zürich, Switzerland
[2]Swiss Finance Institute, c/o University of Geneva, Geneva, Switzerland

JCG, 0000-0003-3219-1588

We present a detailed bubble analysis of the Bitcoin to US Dollar price dynamics from January 2012 to February 2018. We introduce a robust automatic peak detection method that classifies price time series into periods of uninterrupted market growth (drawups) and regimes of uninterrupted market decrease (drawdowns). In combination with the *Lagrange Regularization Method* for detecting the beginning of a new market regime, we identify three major peaks and 10 additional smaller peaks, that have punctuated the dynamics of Bitcoin price during the analysed time period. We explain this classification of long and short bubbles by a number of quantitative metrics and graphs to understand the main socio-economic drivers behind the ascent of Bitcoin over this period. Then, a detailed analysis of the growing risks associated with the three long bubbles using the *Log-Periodic Power-Law Singularity* (LPPLS) model is based on the *LPPLS Confidence Indicators*, defined as the fraction of qualified fits of the LPPLS model over multiple time windows. Furthermore, for various fictitious 'present' times $t_2$ before the crashes, we employ a clustering method to group the predicted critical times $t_c$ of the LPPLS fits over different time scales, where $t_c$ is the most probable time for the ending of the bubble. Each cluster is proposed as a plausible scenario for the subsequent Bitcoin price evolution. We present these predictions for the three long bubbles and the four short bubbles that our time scale of analysis was able to resolve. Overall, our predictive scheme provides useful information to warn of an imminent crash risk.

# 1. Introduction

From an investment point of view, during the past decade, Bitcoin has become known for two main reasons: its extraordinary return potential in phases of extreme price growth as well as regular

massive crashes of the cryptocurrency. For instance, as a consequence of the crash following mid-December 2017, a book-to-market value of more than 200 billion US Dollars of Bitcoin's total market capitalization evaporated within only six weeks, resulting in a cumulative loss from the peak of 41% (over 42 trading days after the peak that occurred in mid-December 2017). The massive crash was preceded by a no less impressive 43 times price boost (over 730 days before the peak in mid-December 2017). Introduced in 2008 [1], Bitcoin started trading on organized markets in 2010 and, from the beginning, has exhibited a turbulent market history such that the bubble culminating in December 2017 does not appear to be that exceptional. In fact, as we will demonstrate in this paper, multiple overlapping short- and long-term Bitcoin price bubbles have appeared between 2012 and 2018. The goal of this study is to document these bubbles and crashes, put them into a historical perspective and analyse their predictability.

At the time of writing, the combined capitalization of all existing cryptocurrencies still amounts to less than 1% of the world GDP [2], a fact illustrating the still low significance of this market in the global economic context. Nevertheless, cryptocurrencies, and especially Bitcoin as a precursor of this new asset class, have drawn increased attention by the scientific and investor communities, due to the strong growth of the sector over the past years as well as the promising technological and economic prospects.

There have already been a number of studies examining the statistical properties of Bitcoin returns. Pichl & Kaizoji [3] modelled the time-varying realized volatility of Bitcoin and found it to be significantly larger compared to that of fiat currencies. Urquhart & Zhang [4] studied a variety of GARCH volatility models and tested the hedging capability of the crypto-coin against other currencies. The hedging properties against other asset classes were investigated by Bouri *et al.* [5]. Bariviera [6] provided evidence for volatility clustering through a long memory Hurst exponent analysis. Furthermore, Osterrieder & Lorenz [7] found much larger magnitude in the heavy tail of the Bitcoin return distribution compared to conventional currencies. Additionally, Begušić *et al.* [8] determined even larger tail risk than usually seen in stocks. Donier & Bouchaud [9] investigated Bitcoin liquidity based on order book data and, from this, accurately predicted the size of price crashes. Other approaches to econometric Bitcoin modelling are outlined in [10].

Besides interest in the purely statistical properties of the Bitcoin financial time series, there has been growing focus on the social component shaping Bitcoin price dynamics. Kristoufek [11] firstly observed a bidirectional relationship between web search queries and Bitcoin prices. Further on, Garcia *et al.* [12] detected positive feedback loops between Bitcoin prices, user numbers of the Blockchain network and search queries. They successfully implemented a profitable Bitcoin trading strategy exploiting these social dynamics [13]. Likewise, Glaser *et al.* [14] investigated the link between Bitcoin prices, Blockchain network and search query data. They observed that 'Bitcoin users are limited in their level of professionalism and objectivity' because they primarily use Bitcoin as a speculative investment. An analysis of the Bitcoin user base in more depth was carried out by Kim *et al.* [15], who applied topic modelling to Bitcoin forum posts and evaluated the predictive power of a deep learning model that was trained on the forum data. Further contributions to the characterization of the Bitcoin community are also summarized in [10].

These studies of Bitcoin's price and social dynamics suggest that Bitcoin buyers have mainly been attracted by the sky-rocketing price performance of the cryptocurrency and were influenced by news and social media. This is a typical characteristic of previously seen financial bubbles. Unsurprisingly, therefore the media as well as many pundits have drawn parallels between the Bitcoin phenomenon and former extraordinary financial bubbles such as the Tulip Mania [2]. In this study, we present confirming evidence and quantitative analyses that strongly support the conclusion that Bitcoin has behaved as a highly speculative asset exhibiting strong bubble activity.

In order to test for such bubble activity, we use the *Log-Periodic Power-Law Singularity* (LPPLS) model. First introduced as a calibration model by Sornette *et al.* [16], it was elaborated into a rational expectation bubble model in [17,18] to provide real-time diagnostics of bubbles and forecasts of crashes. A few studies have already applied the LPPLS model [19,20] and other econophysics bubble models [21,22] to analyse past Bitcoin bubbles.

Here, we present a comprehensive general methodology to identify and classify the complete set of both short and long Bitcoin bubbles that occurred within the studied time period from 2012 until 2018. The methodology for bubble characterization is completely automatized and can be applied to any asset price time series, making it a robust measure for bubble detection. Here, we base our analysis on daily Bitcoin to US Dollar (*btc/usd*) price data, as quoted on the *Bitstamp* exchange from August 2011 on, when the exchange was founded. The dataset was obtained from Thomson Reuters Datastream [23].

We identify three massive long bubbles and 10 short bubbles. The temporal coexistence of these bubbles underscores the multiscale nature of bubbles. It motivates the development of metrics that

can be used to separately identify evolving short and long bubble dynamics. We explore two techniques for this: (i) the LPPLS Multi-Scale Confidence Indicator [24,25] and (ii) the characterization of bubble burst scenarios obtained by clustering LPPLS calibrations over time scales and predicted critical times. As a core part of this study, the forecast ability of the metrics to diagnose bubbles in real time and predict crashes is evaluated. The present study complements and extends the one presented by Wheatley *et al.* [26], which focused on the identification of a proxy for the fundamental value of Bitcoin and a preliminary assessment of predictability of a subset of the bubbles studied here. Here, we dwell much more in exploiting the information contained in the LPPLS model calibration using the two mentioned sophisticated indicators.

The paper is structured as follows.

In §2, we present our framework for bubble and crash identification. We apply it to the Bitcoin time series, present and quantitatively characterize the set of short and long bubbles that we obtain. Section 3 discusses some of the main socio-economic drivers of the detected long bubbles. Our main results are reported in §4 on the real-time predictability of Bitcoin's bubbles. Section 5 concludes.

# 2. Bitcoin bubbles and crashes

Before performing a predictive bubble analysis on Bitcoin, we are motivated to construct a catalogue of the bubbles that actually occurred. The diagnosed bubbles and crashes will provide the targets for our subsequent predictive analyses.

Identifying a bubble ex-post would seem to be a straightforward job, but it is not. In the broadest sense, a bubble could be defined as a large abnormal price increase, which then bursts in a crash. This intuitive description is, however, quite hard to conceptualize and full of traps, as it requires the implicit definition of both 'abnormal price growth' and 'crash'. In order to measure abnormal price increases, a reference frame or process against which deviations can be gauged, must be defined. However, when employing such a reference process, a bubble may be incorrectly diagnosed due to an untrue underlying benchmark model, an issue that makes the diagnostic of a bubble a joint-hypothesis problem [27]. Similarly, a crash is also not easy to define. It can be vaguely described as a mixture of a large loss over some relatively short duration that seems exceptional compared to the regular asset price movements. In the literature, we find a plethora of such definitions, giving an overall impression of unpleasant arbitrariness.

Thus, in this paper, we construct a framework that provides a clear definition of bubbles as transient super-exponential regimes. Based on this definition of bubbles, our methodology combines several metrics to systematically and automatically identify the start and end of long and short bubbles, the size and duration of the subsequent crashes, as well as quantitative characteristics of the bubbles. In the following chapters, we explain the applied techniques in more detail.

## 2.1. Identification of the peak times of long and short bubbles

First, we focus on identifying the peaks of potential bubbles. The peak detection algorithm is based on an extension of the *Epsilon Drawdown Method* developed by Johansen & Sornette [28,29] and further used in [30,31]. In the following, the working principle of the method is summarized. For more details, we refer the reader to the precise explanation of the procedure given in appendix A.

The key purpose of the $\varepsilon$-drawdown procedure is the systematic segmentation of a price trajectory into a sequence of alternating, consecutive price drawup and drawdown phases that are then interpreted as bubbles and crashes. We define a drawup (drawdown) as a period during which the price of an asset generally trends upwards (downwards), but may experience intermittent small movements into the opposite direction. Following this idea, a drawup is defined as a succession of positive returns that may only be interrupted by negative returns no larger in amplitude than a pre-specified tolerance level $\varepsilon$. Likewise, a drawdown is defined as a succession of negative returns that may only be interrupted by positive returns no larger in amplitude than the pre-specified tolerance level $\varepsilon$. Consequentially, a drawup (respectively, drawdown) ends when a negative (respectively, positive) return, whose amplitude exceeds $\varepsilon$, is observed.

By adjusting the parameter $\varepsilon$, we can control the degree to which counter-movements in a drawup or drawdown phase are tolerated. A high value allows for larger oppositely directed movements than a small value, leading to longer drawup and drawdown phases. Ideally, we would like to adaptively adjust $\varepsilon$ depending on the movements of the underlying price series. Therefore, we express it as $\varepsilon = \varepsilon_0 \sigma(w)$, that is the product of a constant, preset multiplier $\varepsilon_0$ and the realized volatility $\sigma(w)$, estimated

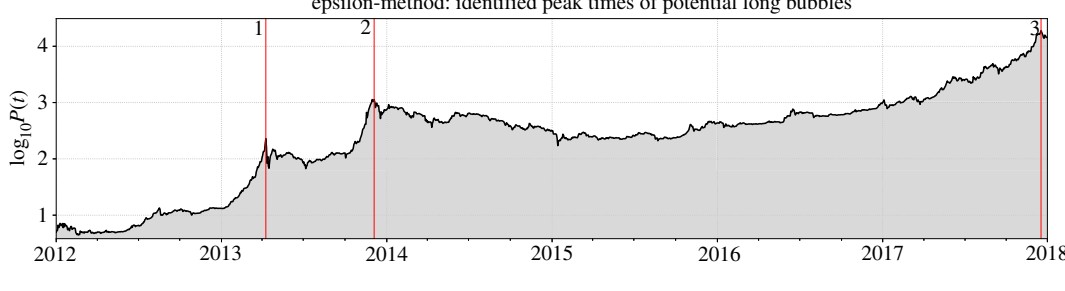

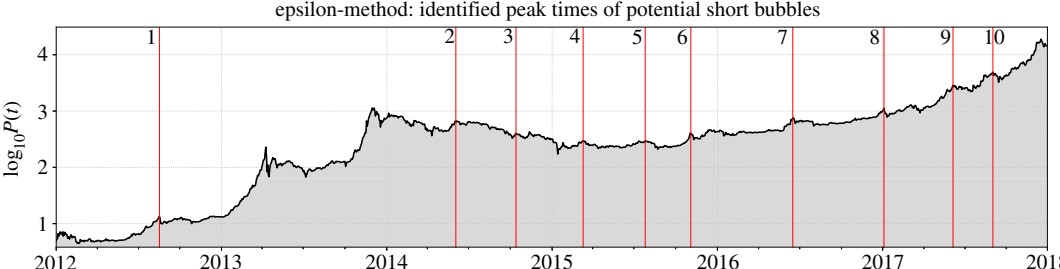

**Figure 1.** Bitcoin bubble peaks identified by application of the generalized *Epsilon Drawdown Procedure* between 2012 and 2018. Depicted in both panels is the logarithm of the *btc/usd* exchange rate (black line, grey region) from January 2012 to January 2018. The vertical red lines indicate the bubble peaks that were identified according to the procedure outlined in the text. (*a*) The three vertical lines indicate the peaks obtained with the condition $0.95 \leq f_t \leq 1$ (see text) and can be interpreted as the peak times of long bubbles. (*b*) The 10 vertical lines indicate the 10 peaks selected with the condition $0.65 \leq f_t < 0.95$. They can be interpreted as the peaks of short bubbles.

over a running window of the past $w$ days. $\varepsilon_0$ controls the number of standard deviations up to which counter-movements are tolerated, and $w$ determines the time scale on which the standard deviation is estimated. In this way, we enable the procedure to be more forgiving during times of hectic market activity with respect to oppositely directed movements, while closing in more strictly on the price trend in times of low volatility.

As pointed out, the outcome of the $\varepsilon$-procedure is dependent on the choice of the pair ($\varepsilon_0$, $w$). Rather than setting specific values of these variables, for the sake of robustness, we perform a grid search over a predefined search space of ($\varepsilon_0$, $w$)-pairs. More precisely, we scan $\varepsilon_0$ from 0.1 to 5 in steps of 0.1 and, for each value of $\varepsilon_0$, we scan $w$ from 10 to 60 days in steps of 5 days. This means that we allow for up to a five standard deviation return to interrupt a current trend and measure the standard deviation over a moving window size of up to two months.

The described choice of pairs yields in total $50 \times 11 = 550$ grid points ($\varepsilon_0$, $w$). For each pair, we run the segmentation procedure over the *btc/usd* time series between January 2012 and January 2018, and obtain a sequence of drawups and drawdowns that is unique for each pair ($\varepsilon_0$, $w$). From each raw sequence, we record a set of peak times $\{t_p\}$. These peak times are defined as the end dates of all drawup periods that are contained in the sequence. The underlying thought is that, if drawups are seen as bubble phases, then their ends probably correspond to bubble peaks.

Scanning over all 550 pairs ($\varepsilon_0$, $w$) gives us 550 sets of peak times $\{t_p\}$. Now, for each daily date $t$ in the observed timeframe, we count the number of times that this date was identified as a peak time over the whole set of 550 pairs ($\varepsilon_0$, $w$). Dividing the resulting number of counts by 550 gives us the fraction $f_t$ of pairs ($\varepsilon_0$, $w$) that have the date $t$ qualified as a drawup peak.

Ultimately, we would like to classify the identified peaks into short bubble or long bubble peaks. Clearly, long bubbles correspond to larger price movements because long bubbles have more time to grow bigger. Therefore, their peaks are likely to be identified by the $\varepsilon$-method on most observed pairs of ($\varepsilon_0$, $w$). Thus, we define large peaks, interpreted as the end of long bubbles, as those dates such that $0.95 \leq f_t \leq 1$, i.e. as bubble peaks that are identified more than 95% of the time. Short bubbles, on the other hand, correspond to relatively smaller price movements. Therefore, we expect to identify them less often than long bubbles. For this reason, we define peaks of intermediate sizes, interpreted as the end of short bubbles, as those dates such that $0.65 \leq f_t < 0.95$.

Figure 1 shows the results of the application of our procedure. The upper frame shows three vertical lines identifying the three peaks with the condition $0.95 \leq f_t \leq 1$. These three peak times are pleasantly coincident with the ends of the three largest bubble regimes of the Bitcoin price in US Dollars from

January 2012 to February 2018. By the time of finalizing the reviewed version of this paper (February 2019), in hindsight, it is also evident that the peak of the third bubble in 2017 corresponds to a true bubble peak. The lower frame shows 10 vertical lines identifying the 10 peaks, other than the three previous peaks, selected with the condition $0.65 \leq f_t < 0.95$. These 10 peak times will be interpreted below as the ends of shorter bubble regimes. Note that these sets of peak times are robust with respect to significant (more than $\pm 20\%$) changes of the thresholds 0.95 and 0.65.

## 2.2. Identification of the beginning times of long and short bubbles

Following the above systematic determination of the main bubble peak times, we next need an automatic, unbiased determination of the beginning times of the corresponding bubble regimes. For this, we use the LPPLS model, first introduced by Sornette [16] and elaborated upon by Sornette & Johansen [17,18]. The model combines a description of the faster-than-exponential transient price acceleration with an increasing frequency of volatility fluctuations that are deemed characteristic of a bubble regime. A summary of how the LPPLS model derives from the Johansen–Ledoit–Sornette model [17,18] is given in appendix B. The LPPLS formula reads

$$\text{LPPLS}(\boldsymbol{\phi}, t) := E[\ln p(t)] = A + (t_c - t)^m (B + C_1 \cos(\omega \ln(t_c - t)) + C_2 \sin(\omega \ln(t_c - t))). \quad (2.1)$$

The model includes seven parameters $\boldsymbol{\phi} = \phi_1 \cup \phi_2$, of which four are linear $\phi_1 = \{A, B, C_1, C_2\}$ and three are nonlinear $\phi_2 = \{m, \omega, t_c\}$. The calibration methodology of the LPPLS model to a given price time series is explained in appendix C. In a nutshell, we apply a two-step procedure. For a fixed value of the triplet $(t_c, m, \omega)$, we solve analytically for the four linear parameters, which are solution of a linear matrix equation deriving from the ordinary least squares formulation. We perform a grid search in the $(t_c, m, \omega)$-domain, followed by a nonlinear minimization, with the conditions that $t_c - t_2$ lies between 0 and $t_2 - t_1$, $0 < m < 1$ and $1 < \omega < 50$.

Fitting the LPPLS formula to the *btc/usd* log-price trajectory requires the choice of a specific time window $[t_1, t_2]$. The window end time $t_2$ can be placed directly at the bubble peak for post-mortem analyses. In order to diagnose and eventually forecast the future price evolution, in an ex-ante predictive framework, $t_2$ would represent the 'present time', up to which the price history is available for analysis. Choosing the beginning time $t_1$ (or equivalently the window size $dt := t_2 - t_1 + 1$) is less straightforward. Optimally, one would like to select $t_1$ as the beginning of the current bubble (or later) if a bubble is developing, such that non-bubble phases, i.e. phases of regular, exponential price growth, are separated from the data in the fit window.

To determine the beginning of a bubble, we implement the Lagrange regularization approach introduced by Demos & Sornette [32]. For a fixed $t_2$, we scan $t_1$ from $t_2 - 29$ to $t_2 - 719$, corresponding to 691 windows ranging from 30 to 720 days. In principle, taking into account all LPPLS fits over the 691 time windows, the beginning of the bubble is defined as the time $t_1^*$ that gives the 'best'-fit quality. Naively, one could use the square root of the normalized sum of squared errors, also known as the root-mean-square (RMS), and choose $t_1^*$ as the value that minimizes the RMS. However, this procedure is incorrect as it does not account for the fact that smaller windows will be favoured, due to their smaller number of degrees of freedom. Demos & Sornette [32] observed that applying a simple correction to account for this bias, which is linear in the window size $(t_2 - t_1)$, behaves quite well and was efficient in a number of tests and real-life case studies. This *Lagrange Regularization Approach* is recalled in appendix D. We apply the method to the set of LPPLS fits calculated at each of the obtained short and long bubble peaks on the *btc/usd* rate in the same timeframe as before. We impose the constraint that, for a given developing bubble, its start time $t_1^*$ cannot be earlier than the previous peak, as determined in figure 1.

## 2.3. List and main characteristics of the long and short bubbles

Combining the bubble start and end times that were determined in the previous subsections, we obtain a set of bubbles encoded by their time interval $[t_1^*, t_p]$, i.e. the timeframe ranging from start to peak of the bubble. By definition, after the peak time of a bubble, a drawdown starts, which initiates the crash or correction regime. We define the end of this correction regime as the time at which the price reaches its minimum value over the time interval starting from the beginning of the drawdown up to the start time of the next identified bubble. For the last analysed bubble, we simply take the price minimum between the bubble peak and the last available data point. The intermediate regime separating the previous crash end and the next bubble start, if there exists one, is then defined as a phase of non-bubble price growth.

**Table 1.** Basic information about all qualified bubbles. Dates and characteristics defining each bubble cycle. For each bubble, we provide the associated timeframe consisting of the bubble start date $t_1^*$, its peak date $t_{peak}$ and the crash end date $t_{ce}$. Crash start dates, which are not listed here, by definition occur one day after the individual peak times. Furthermore, as an absolute measure, Bitcoin prices in US Dollars (rounded to integer numbers) at the individual peak and crash end dates are quoted. The respective durations of the bubbles in days, as well as the bubble and crash sizes (in %) are given in the next three columns. As described in the text, these properties were calculated based on the previously identified bubble timeframes. Moreover, they were calculated based on exact prices (and not based on the rounded ones that are given in this table). The last column in the short bubble part of the table indicates whether a potential bubble passed (Y) or did not pass (N) the bubble filtering conditions defined in the main text.

| no. | start $t_1^*$ | peak $t_{peak}$ | crash end $t_{ce}$ | $P[t_{peak}]$ | $P[t_{ce}]$ | duration (days) | bubble size (%) | crash size (%) | |
|---|---|---|---|---|---|---|---|---|---|
| long bubble data | | | | | | | | | |
| 1 | 28 May 2012 | 9 Apr 2013 | 16 Apr 2013 | 229 | 68 | 316 | 4416 | −70.27 | |
| 2 | 3 July 2013 | 4 Dec 2013 | 14 Jan 2015 | 1132 | 172 | 154 | 1367 | −84.83 | |
| 3 | 15 Jan 2016 | 18 Dec 2017 | 25 Dec 2017 | 18941 | 13911 | 703 | 5152 | −26.55 | |
| short bubble data | | | | | | | | | |
| 1 | 7 May 2012 | 16 Aug 2012 | 20 Aug 2012 | 13.4 | 10.1 | 101 | 165 | −25.11 | Y |
| 2 | 15 Apr 2014 | 3 June 2014 | 25 June 2014 | 670 | 560 | 49 | 28 | −16.38 | Y |
| 3 | 30 June 2014 | 14 Oct 2014 | 23 Oct 2014 | 403 | 357 | 106 | − 37 | −11.45 | N |
| 4 | 23 Oct 2014 | 11 Mar 2015 | 13 Apr 2015 | 297 | 223 | 139 | − 16 | −24.75 | N |
| 5 | 13 Apr 2015 | 27 Jul 2015 | 24 Aug 2015 | 294 | 210 | 105 | 31 | −28.74 | Y |
| 6 | 26 Aug 2015 | 4 Nov 2015 | 11 Nov 2015 | 408 | 310 | 70 | 80 | −24.04 | Y |
| 7 | 15 Jan 2016 | 16 June 2016 | 2 Aug 2016 | 767 | 540 | 153 | 112 | −29.56 | Y |
| 8 | 3 Aug 2016 | 4 Jan 2017 | 11 Jan 2017 | 1115 | 779 | 154 | 97 | −30.16 | Y |
| 9 | 24 Mar 2017 | 6 June 2017 | 7 June 2017 | 2881 | 2683 | 74 | 210 | −6.86 | Y |
| 10 | 7 June 2017 | 1 Sep 2017 | 14 Sep 2017 | 4922 | 3228 | 86 | 83 | −34.42 | Y |

From the knowledge of start, peak and crash end dates of each bubble, as well as the price trajectory, we calculate the bubble size in per cent, defined as the cumulative return between bubble start and peak for each bubble. Additionally, the bubble duration is computed as the number of days between bubble start and peak. Ultimately, the crash size is calculated as the cumulative return between the peak time and crash end date. We thus have the systematically determined beginning time and other characteristics, which are summarized in table 1.

For the 10 short bubble candidates associated with the peaks shown in the lower panel of figure 1, we want to avoid very short durations and small sizes and thus impose two additional filters: a 'short' bubble is such that it duration is at the minimum 30 days and its size larger than 25%, i.e. a short bubble has its price increase by at least 25% over at least 30 days. After applying these filtering conditions, for the set of 10 short bubble peak times shown in the bottom frame of figure 1, we find that eight of them qualify as end times of real short bubbles. Two of the 10 peaks (Peaks 3 and 4) are excluded as not being preceded by a sufficiently large price increase (actually, they are local peaks within or just after a very low drawdown shown in figure 2a).

Figure 2 shows the three identified long (figure 2a) and eight short (figure 2b) bubbles. At a first coarse-grained level of description, the top panel suggests that the Bitcoin history can be divided into three main regimes: (i) a pre-2014 phase of intense price growth, (ii) a 2014–2016 long drawdown and side-way regime, followed by (iii) the massive 2-year-long bubble that recently burst at the end of 2017. The bottom panel shows that the two large-scale bubbles from 2012 to end of 2013 and from 2016 to end of 2017 are themselves composed of shorter bubbles, some of them still with impressive amplitudes.

## 2.4. Additional robustness check by change point detection

The bubble classification presented in the previous subsection will serve as the input for subsequent analyses. In this section, in order to check further the reliability of our preceding bubble detection

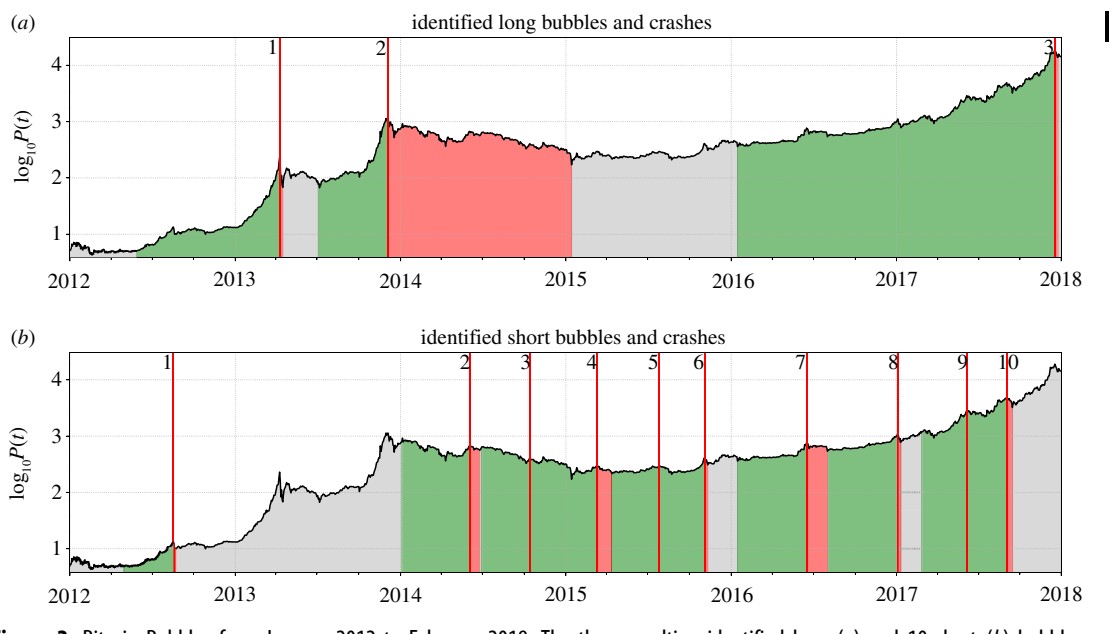

**Figure 2.** Bitcoin Bubbles from January 2012 to February 2018. The three resulting identified long (*a*) and 10 short (*b*) bubbles, before filtering for short bubble characteristics. The green bands delineate the bubble growth phases, while the red shaded regions represent the associated crashes or correction regimes. The grey areas indicate phases of 'non-bubble price growth', as defined in the text. The numbers to the left of the vertical red lines map to those given in figure 1. We can see that Short Bubbles 3 and 4 do not qualify as real bubbles, as their return over the green growth phase is negative (table 1). As stated in the text, they were therefore removed from the analysis.

procedure, we present the application of a standard change point detection algorithm to the return series of Bitcoin. This reveals regimes in Bitcoin returns that are similar to those identified by our specialized method, confirming the robustness of our bubble identification results.

Figure 3 depicts the log-return series of Bitcoin during the analysed time period from 2012 to 2018. On top of the returns, the 30-day running window mean (red line) and standard deviation bounds (grey shaded area) are plotted. Significant changes in both the mean and the standard deviation over time are observable. The mean temporarily deviates from zero especially during the phases that we pointed out as the long bubble periods in the main text. In addition, the highly non-stationary standard deviation indicates increases in volatility in response to the observed crashes.

Following this simple visual representation of non-stationarity of the Bitcoin time series around the bubble phases that we identified, we turn to the change point detection. We use the open source Python module `ruptures` [33] that lets us apply a variety of algorithms to search structural breakpoints in the time series. We use the implemented version of the Power of the Pruned Exact Linear Time (PELT) algorithm [34] that allows for the detection of a previously unspecified, variable number of change points. The results of the procedure are depicted in figure 4. Very similar results are obtained with other change point detection algorithms and settings.

The PELT method identifies changes in the return behaviour around the growth and peak phases of the previously identified long bubbles in 2013 and 2014. Furthermore, it finds a similar duration for the long recession between 2014 and (here the first half of) 2015. For the 2017 bubble, a structural change around the beginning of 2017 is detected, which is about 1 year after the bubble start diagnosed with our more specialized method.

Overall, using the PELT algorithm, one can observe similar major regime changes as the ones that we have previously identified with our specialized method. Given that it is designed specifically for the detection of drawdowns and drawups in financial time series, our bubble detection framework provides a much more accurate determination of the exact regime transitions, which correspond to the start and peak times of bubbles. We argue that our technique is more adapted to financial markets, because it focuses on the financially relevant concept of drawdowns and drawups, while the PELT method is more general, and thus *a priori* less powerful for the specific financial application.

We now turn to the description of the three main long bubbles that were presented in the previous subsection.

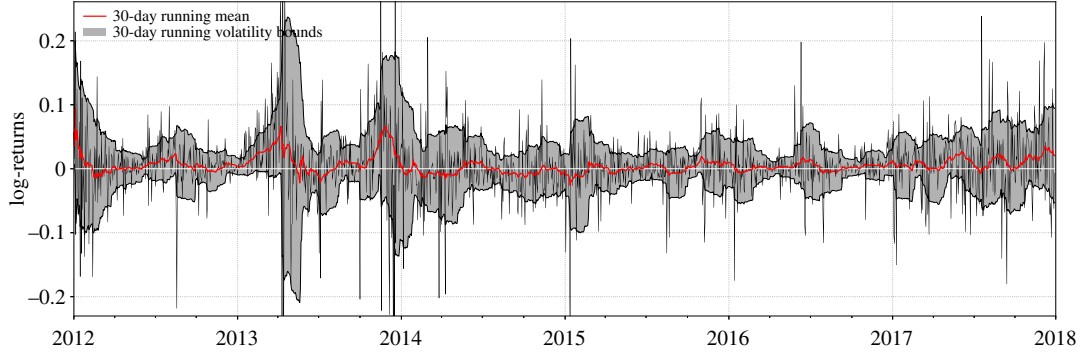

**Figure 3.** Running window analysis of mean and variance of Bitcoin log-returns. This simple analysis shows non-stationary behaviour in both mean and variance of the log-return series of Bitcoin.

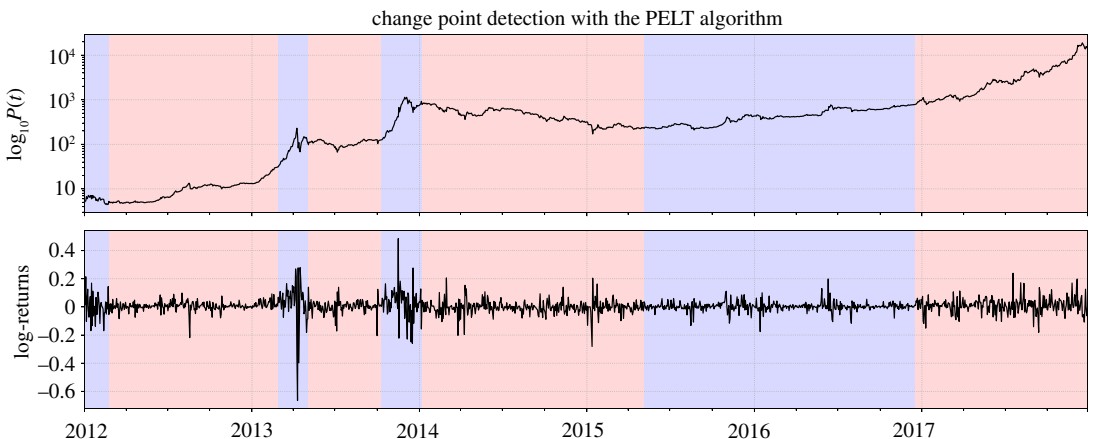

**Figure 4.** Change point detection. Application of the PELT algorithm reveals seven structural change points in the *btc/usd* log-return time series that are indicated by the transition between red and blue shaded regions. Note that, here, the red and blue colours simply serve to distinguish the resulting different regimes, not to classify the time series into bubble or crash periods.

# 3. Socio-economic drivers behind Bitcoin bubble dynamics

Having characterized the main bubble events, before quantitatively studying their predictability in §4, it is useful to put these events into context and expose their key drivers, as well as the developments and events that may have promoted their nucleation or have caused their sudden crashes. Many previous works have shown that bubbles usually grow out of a rational reaction to a change in economic conditions, or to novel opportunities, new technologies, and so on [35–39]. Then, through positive feedback processes, the price dynamics amplify beyond what seems justifiable. In this spirit, it is the goal of this chapter to focus on what could have been such novel pieces of information that may have nucleated the bubbles. We are well aware that correlation is not causation and our discussion here is more qualitative, with the goal of offering a partial account of the atmosphere in which the identified bubbles developed. This section is thus more descriptive and sets up the backdrop against which to interpret the quantitative findings of the next section.

In particular, we analyse the possible causes of the nucleation and main drivers for the growth of the three major identified Bitcoin long bubbles that occurred between 2012 and 2018. The history of Bitcoin at its early stage was highly influenced by fiscal and monetary measures undertaken during the Eurozone crisis, as will be elaborated below. Later on, the development of the cryptocurrency was to a great extent influenced by increasing demand from China, which evolved as a result of the emergence of major Chinese Bitcoin exchanges opening up the markets for investors. Although it had a strong temporary impact on Bitcoin in the short term, the shutdown of Chinese exchanges that occurred in early 2017 caused no persistent loss in its capitalization. Quite on the contrary, the year of 2017 is characterized by remarkable growth dynamics due to the contagion of the Bitcoin bubble to a general cryptocurrency bubble. Finally, at the end of 2017, the most recent long bubble crashed with a shocking

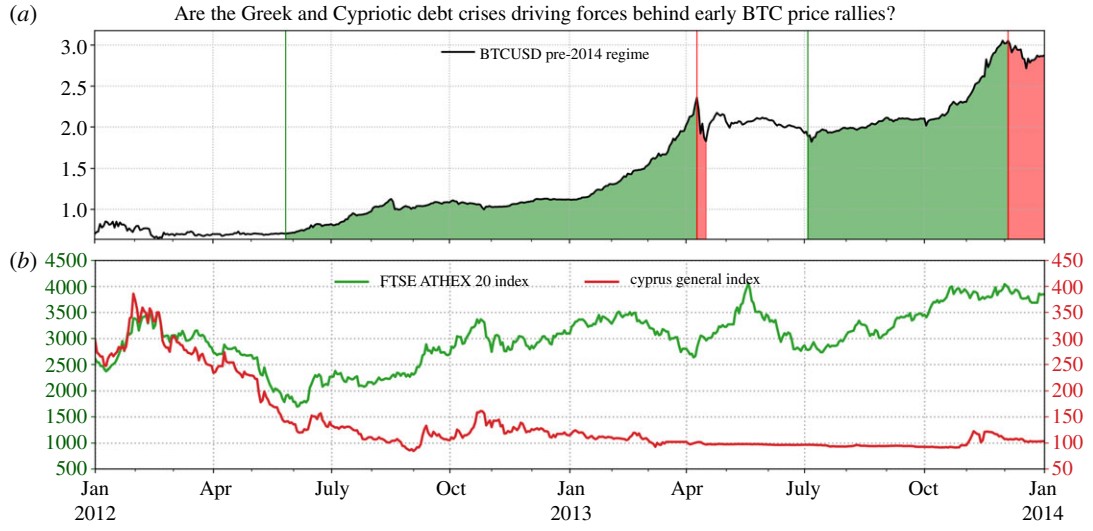

**Figure 5.** Comparison of the evolution of Bitcoin Price and Greece as well as Cypriot Financial Market Indices. (*a*) The Bitcoin log-price trajectory from January 2012 to January 2014. (*b*) The Greek ATHEX 20 Index (left scale), as well as the Cyprus General Index (right scale) are depicted over the same period. The financial market indices of Spain, Portugal and Ireland follow paths highly correlated with that of the Greece index, as their indices bottomed out during approximately the same time. (Data Source: [23].)

intensity, removing 60% (up to February 2018) of the cryptocurrency market capitalization measured from its peak in December 2017. The following subsections examine this history in more details.

## 3.1. First long bubble: May 2012–April 2013

The financial crisis of 2007–2008 put economies worldwide into a fragile state. It also helped reveal the unsustainable debt level and the longstanding financial mismanagement of some small- and medium-sized economies within the European Union such as Greece, Ireland, Portugal, Spain, Iceland and Cyprus. During the European Debt Crisis, or Eurozone Crisis, the crises in Cyprus and Greece stood out by their intensity and consequences, as subsequent developments demonstrated. Notice that both countries reached their worst economic state for decades in 2012, as demonstrated in figure 5*b*. The economic decline was accompanied by ongoing discussions about further bailout programmes for Greece, as well as requests of the Cypriot government for a bailout of its financial sector which the country's economic well-being largely relied on [40].

The emerging mentality of distrust in governments and financial institutions triggered a wave of bank runs and hunts for monetary safe havens. Bitcoin, proposed as an alternative store of value that was primarily intended to be uncontrollable by governments and independent of monetary policies [1], appeared to ideally meet these requirements. It is therefore interesting to observe that the nucleation of the first Bitcoin long bubble occurred at the exact time when the Greece and Cypriot indices reached local troughs. Additionally, the price increase in March–April 2013 may have been driven by prominent Silicon-Valley-based investors [41]. It is even possible that both factors are connected, the savvy investors recognizing the implications of the Cyprus-Greek crisis and betting on it by bidding on Bitcoin. Market manipulation, whose presence has been revealed more recently [42,43], may also have played an important role, for this bubble as well as in the later bubble episodes.

On 16 March 2013, a critical point was reached for the Cypriot economy, when a bail-in tax that largely affected the deposits of account holders at Cypriot banks was declared. The announcement created a massive wave of bank runs of account holders trying to protect their personal savings [44]. The event coincided with the last accelerating growth of Bitcoin's price, before its crash in April 2013 [45].

The described developments in Cyprus were anxiously observed in other Eurozone countries, for instance, in Spain, where people feared that similar governmental interventions could lead to the loss of their own savings [46]. Serving as a store of value that could not be seized by any institution, Bitcoin arrived on the scene at the right time, perceived as the perfect alternative investment to hedge against monetary interventions. Figure 6 illustrates the consequential flow towards the cryptocurrency in form of soaring Blockchain transactions around the time of the bubble nucleation of the first long bubble. As a response to the increasing demand, in the course of bubble growth, the price of a Bitcoin was catapulted up by an incredible 4400% above the price mark of 100 USD per Bitcoin.

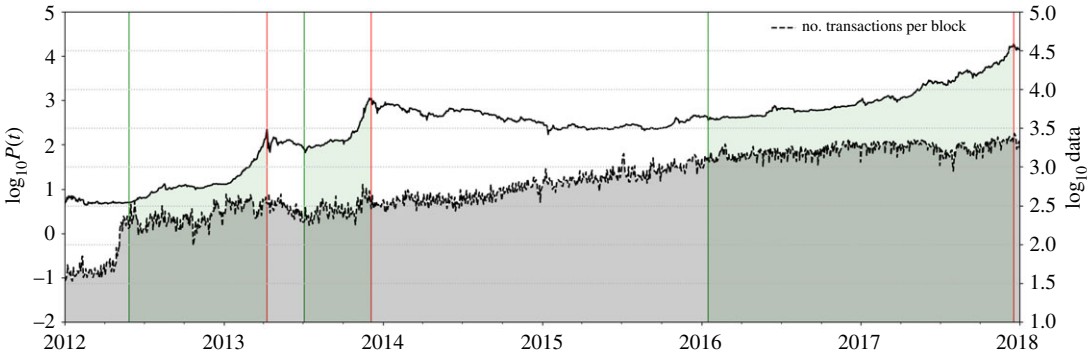

**Figure 6.** Evolution of the number of transactions on the Blockchain network. The grey shaded region delineates the logarithmic number of Bitcoin Blockchain transactions (right scale). In 2012, a jump of the transaction number occurred in response to the large increase in demand by investors seeking Bitcoin as a safe haven. (Data Source: [47].)

The bubble reached maturity in early April 2013. After peaking on 9 April, it burst, with approximately 70% of Bitcoin's market capitalization disappearing within one week. The actual cause of the crash was a reaction to instability announcements on *MtGox*, the by-then worldwide biggest Bitcoin Exchange in terms of Bitcoin exchange-traded volume, which was suffering the consequences of a hacker attack [48].

Obviously, the event leading to the crash was not directly related to the political backgrounds of the Eurozone and Cypriot crises that we, however, identified as likely driving factors in the formation of this bubble, as stated above. As is often the case in the burst of a bubble, one should distinguish the cause of the crash, which is in general unrelated to the source of the bubble, from the fundamental origin of the crash [36,49], which is that the Bitcoin market had progressively evolved towards a fragile, unstable, critical state, associated with a large susceptibility to adverse news.

## 3.2. Second long bubble: July 2013–December 2013

The second long bubble matured at the end of 2013 after an approximate 13-fold increase from its post-crash recovery level after the first long bubble. As the European debt crisis was still well underway during this time, again, the attraction to the alternative decentralized cryptocurrency Bitcoin contributed to its price surge. But one should not underestimate a number of additional factors that contributed to the emergence of this bubble: the adoption of Bitcoin in China, the FBI shutdown of the Darknet drug market *Silk Road*, as well as the growth and increasing technical sophistication of the Bitcoin mining community.

Figure 7 presents the history of the birth and trading volumes of the main Bitcoin exchanges from 2012 onward. The blue inset plots show the births and absolute volumes handled by the different exchanges. The light grey area shows the logarithmic summed transaction volume of all of these exchanges that are still active today, while the dark area sums the volume of all listed exchanges. It can be seen that major Chinese exchanges were founded in the nucleation phase of the second long bubble, among them *Huobi* and *OKCoin*, which later on became the dominating Bitcoin exchanges in China [51].

The emergence of new Bitcoin exchanges significantly facilitated market entrance for the numerous primarily China-based investors to the Bitcoin market. Ultimately, the wave of new investments drove prices above the 1000 USD level for the first time. Figure 8 shows that the share in total traded volume of Chinese exchanges started to really break through during the second long bubble, in the phase of most intense growth. Before that, *MtGox* held the majority of the market share. However, its market share plummeted from more than 50% at the beginning of the bubble down to a mere 10% at its peak. The corresponding lost fraction had been replaced by trading volume on the uprising Chinese exchanges.

As the bubble was developing at fast pace, on 2 October 2013, the online drug market *Silk Road*, which enabled customers to buy illegal substances anonymously using Bitcoin, was shut down by the FBI and its alleged founder imprisoned [52]. This signalled to the Bitcoin community that the legal authorities had their eye on the cryptocurrency and intended to prevent any illegal activities related to it by all means. *Silk Road* was not the only, but by far the most popular Darknet drug market at the time. Therefore, its seizure had wide implications and hit the news headlines heavily. The closure of *Silk Road* symbolically set free Bitcoin as a proper investment for more cautious investors who, until then,

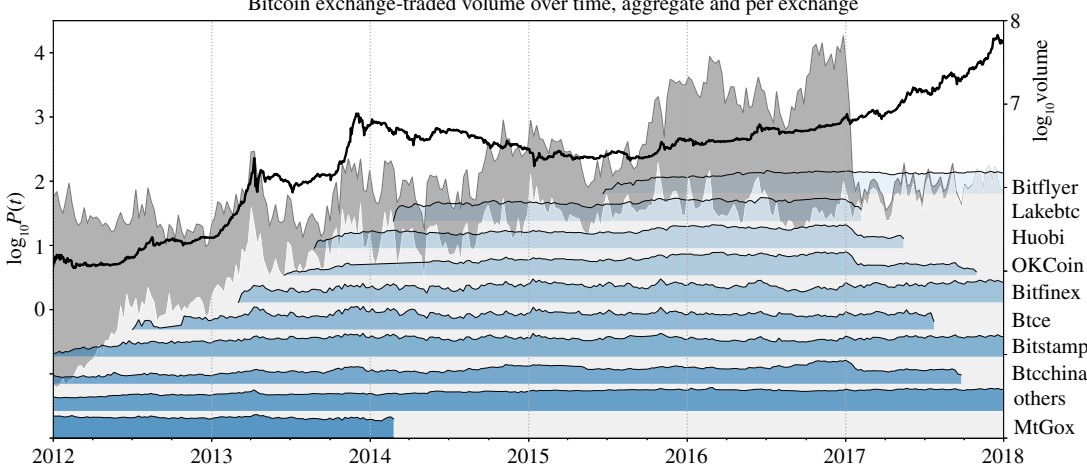

**Figure 7.** The evolution of Bitcoin exchanges in terms of trading volume. The dark grey area (right scale) represents the total log-volume of all analysed exchanges. The light area shows the volume of only the exchanges that are still in business as of end 2018. The blue inset plots schematically show the evolution of traded log-volume of the single exchanges that are listed next to the right plot axis. Note that the inset plots are ordered according to the dates of entry into service of the different exchanges. Hence, from these plots, the temporal origination of exchanges can be compared. (Data Source: [50].)

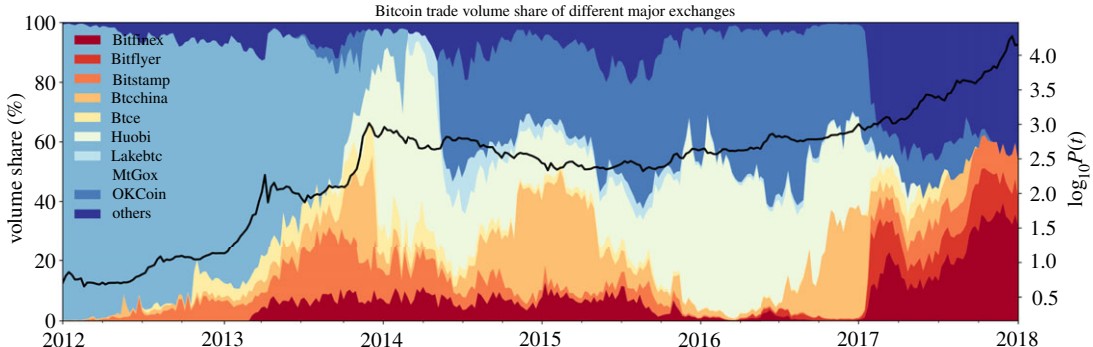

**Figure 8.** Volume share of the top 10 Bitcoin exchanges. Shown on the left scale is the percentage share of the Bitcoin volume traded on each exchange with respect to total traded volume. Notice the surge of the two main Chinese exchanges, *OKCoin* and *Huobi* towards the end of the second long bubble, as well as the constant decrease of the market share of until then dominating Japan-based exchange *MtGox*. (Data Source: [50].)

were deterred by its illegal usage as drug money. The subsequent build-up of a clean image for Bitcoin was also recognized by the US senate, which as a consequence announced an official hearing about the utility and future prospects of the digital currency [53]. The event is seen as a further beneficial factor contributing to the immense price surge seen in the second long bubble [54].

Figure 9 hints at the third contributing factor for the price growth during the bubble. The logarithmic hash rate power of the miners computing transaction blocks on the Blockchain is shown, as well as the logarithmic number of registered wallets. Notice that the hash rate grew at a much faster rate compared with the number of wallets over the period starting with the first long bubble. The highest rate of increase was reached during the second half of 2013, coinciding with the second long bubble.

The number of registered wallets is a measure for the size of the network of users operating directly on the Blockchain. Hence, the hash rate increasing faster than the user base signals that on average miners enhanced their individual mining power during the period. This was most likely achieved through the usage of more efficient technical mining hardware. We conclude that, in addition to Bitcoin adoption effects in China and the image improvement of Bitcoin, we can see the effect of the ramp-up behaviour of hash power as an indication of increased mining sophistication.

Concerning the improvement in hardware and mining technology, we note that the first long bubble played an important role as a precursor to the second one. As the Bitcoin price increased for the first time to a fairly high level, miners were incentivized to invest in mining hardware. At the relatively high price

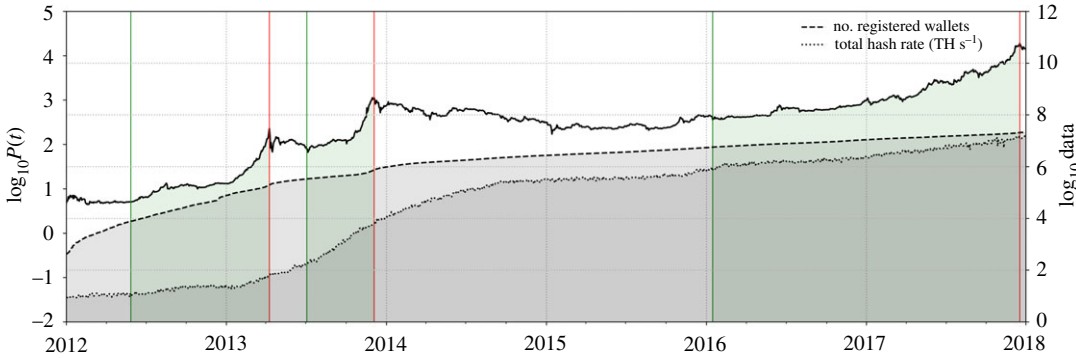

**Figure 9.** Growth of Bitcoin Blockchain network user base and increase in mining power. Superimposed to the Bitcoin price in log-scale (left axis), the total hash rate and the number of registered wallets are shown (right axis). The largest rate of increase of the total hash rate can be observed over the course of the second long-term bubble in 2013, a fact signalling an increasing level of miners' technique sophistication. A second, more subtle acceleration of the hash rate can be seen around the nucleation of the third long bubble. (Data Source: [47].)

of a Bitcoin compared to the computational effort required to create it, it was profitable for them to enter the mining business. The increasing number of miners itself then triggered a feedback mechanism. It can be seen as a self-fulfilling, self-reinforcing bubble in the following sense: the larger the price, the larger the incentive to invest in hardware computing power; the more mining there is, the more activity there is on the Blockchain that attracts more and more users and buyers, the more the price increases. But, the faster growing variable should quickly become the hashing difficulty. The loop is closed when the incentive of miners to invest in hardware rises again.

There are two major triggers for the crash that started in December 2013 and the following long drawdown period. Firstly, the Chinese government suddenly prohibited financial institutions from using Bitcoin when the price of a Bitcoin was close to the peak of the bubble in December 2013 [55]. The announcement destabilized the currency and sent it into free fall. Secondly, during February 2014, when there was still hope for the price to recover, *MtGox* suspended trading (figure 8) and filed for bankruptcy protection from creditors after a major account robbery of at least 700 000 missing or stolen Bitcoins of customers had been reported [56].

The demise of *MtGox* was perceived as a major setback for the Bitcoin world. Account holders of the exchange lost money that was not or hardly recoverable due to the difficulty of tracing it. Once more, this led to a huge loss of Bitcoin's trustworthiness, which already had been questioned over and over again in the past. These adverse circumstances set the currency off into a strong 1-year-long decline.

The question remains whether these events led to exaggerated responses by the investor community. For instance, Fry & Cheah [57] find evidence for a negative bubble in this period, a hint that the market may have overestimated the influence of the discussed events, leading to a stronger than appropriate phase of depression.

## 3.3. Third long bubble: January 2016–December 2017

The winners of the long Bitcoin price drawdown that took place from 2014 until mid-2015 were clearly the China-based Bitcoin exchanges. At the time of the closure of *MtGox*, they had already occupied the majority (more than 90%) of all exchange-traded volume, as can be seen by looking back at figure 8. Within the 3 years following 2014, Bitcoin volume formation on Chinese exchanges contributed to roughly a 100-fold increase in global volume (figure 7). It is this rising demand for Bitcoin from Chinese markets, originating during that time, which can be seen as a major precipitating factor for the nucleation of the third long-term bubble, whose start we identify around the beginning of 2016. This leaves open the question where this increased demand originated from? We identify the devaluation of the Chinese Yuan as a main promoting factor for the rising interest in cryptocurrencies in China and thereby the formation of the third long bubble.

Figure 10 shows a change of regime in the exchange rate of the Chinese Yuan (CNY) versus the US Dollar from 2014 onward. In August 2015, the depreciation of the Yuan was enforced through a devaluation of the currency by the People's Bank of China, *PBoC*, as seen by the sudden jump in the rate. This devaluation was motivated by the desire to raise the competitiveness of exporting firms. From there on, a continuous weakening of the Renminbi developed until January 2017.

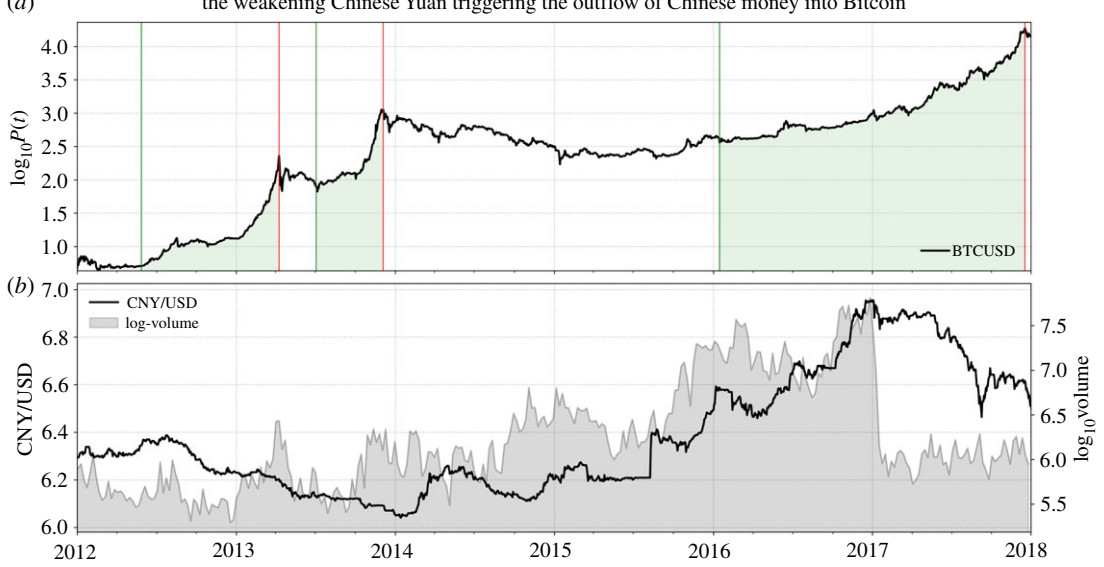

**Figure 10.** Development of Bitcoin in parallel to the Chinese Yuan. The Bitcoin log-price (*a*) and the Chinese Yuan versus US Dollar (CNY/USD) exchange rate (*b*, left axis). A change of regime occurred in the Chinese currency simultaneously to the decline of Chinese exchange-traded Bitcoin volume (*b*, right axis) at the beginning of 2017. (Data Source: [23].)

As a reaction to the depreciation of their currency from 2014 onward, Chinese market participants tried to transfer their money to what they perceived as safer stores of value, causing an outflow of capital from China [58]. As for the average Chinese investor, limitations in terms of foreign-exchange investments were quite restrictive, once more, Bitcoin was a straightforward solution to store value [59]. As a response to the devaluation of the Renminbi, one can observe indeed a large increase in demand for Bitcoin in form of rising trading volume and growing prices from mid-2015 onwards. Note specifically the period of devaluation of the Chinese Yuan (during the last quarter of 2015) preceding the start of the third long bubble.

In January 2017, the Central Bank of China instructed the—until then widely unregulated—Chinese Bitcoin exchanges to comply with the country's financial regulations [60], as it suspected illicit exchange activities such as money laundering, as well as volume manipulation (through wash trading) that was made possible by the zero trading fee policy of exchanges [61]. In a first intervention, the biggest Chinese Bitcoin exchanges *BTCC*, *Huobi* and *OKCoin* [62] were forced to reintroduce a non-zero trading fee structure and to stop leveraged trading [63]. The measure led to the huge slump in exchange trading volume that can be observed in figure 7. Simultaneously, the Short Bubble 8 (figure 2 and table 1) burst in a 30% crash.

Chinese exchanges started with their zero trading fee policy in late 2013. It has been estimated that, due to the ongoing manipulations of trading volume, the volume reported by exchanges had been overstated by up to 40 times the true volume [64]. Hence, the crash in trading volume partly seems to represent a normalization to realistic levels [65], as with the introduction of trading fees, volume could not be generated at no cost, any longer.

In a further regulatory move, in February 2017, the *PBoC* again exerted pressure on Chinese exchanges causing them to halt Bitcoin withdrawals [66], while still tolerating withdrawals executed in the domestic currency Yuan [67]. Effectively, this measure was intended to cut off the outflow of Chinese money through Bitcoin.

In June 2017, the temporary withdrawal pause was ended, as exchanges had partly adapted to the regulatory standards demanded by the *PBoC* [68]. This news implied an overall positive outlook for Bitcoin's future in China, promoting further rise in the price of the digital coin. In September 2017, unexpectedly, Chinese regulators banned the so-called Initial Coin Offerings (ICOs) [69], a novel procedure for the emission of new digital coins, emulating the Initial Public Offerings of regular firms. Finally, in mid-September, the Central Bank ordered Chinese exchanges to shut down all trading activities on the Chinese market [70]. This quick series of unanticipated events officially put an end to cryptocurrency exchange business in China. The gradual decrease in trading volume and closure of exchanges during 2017 is shown again in figure 7.

Although, in late 2017, cryptocurrency trading was proclaimed 'officially dead In China' [71], the actions undertaken by China's Central Bank led to (i) a shift of exchange-based trading to OTC trading, running on

the so-called peer-to-peer exchanges [72], the most known of which was *LocalBitcoins* and (ii) Chinese exchanges such as *Huobi* and *OKCoin* depleting or partly replacing their local exchange business activity while increasing their business activity abroad [73]. In this way, although *PBoC* managed to prevent Yuan-based trading, the still present large if not increased demand was eventually just redirected to other markets [74]. Thus, the imposed regulatory constraints did not have any permanent influence on the global evolution of the price of Bitcoin. This is also backed by the fact that, throughout 2017, the Bitcoin price recovered fast from drawdowns resulting from China-related negative news.

Besides the large contribution of Chinese-based Bitcoin investing to the formation of the third long bubble, there were other factors driving the growth of the bubble. The third long bubble was punctuated by a number of short bubbles that burst abruptly, but were followed by a fast recovery of the Bitcoin price, as seen in figure 2. There are several reasons for this behaviour.

Going back to the nucleation of the third bubble in early 2016, as in the case of the second long bubble, one witnesses an increase in the growth rate of the hash rate, with the total hash power then exceeding 1 ExaHash/second ($=10^{18}H/s$). Moreover, during 2016, several of the nowadays dominating BTC mining pools became active (*viabtc*, *btc.com*, *btc.top*) [75–77]. This development produced additional publicity for Bitcoin and pushed mining efficiency for individuals once more. The fact that mining gains of pools are proportionally shared among miners according to their power contribution provided a large incentive especially for weak miners to resume their mining activity, as their chance of being rewarded for mining (within their lifetime) suddenly was ensured. The growth of the mining network and associated hash power is a factor that may well have contributed to higher resilience and fast recovery of the Bitcoin price during the massive, third long bubble.

The most distinct feature driving the acceleration of the Bitcoin price after the China exchange shutdown was the growth of the cryptocurrency market as a whole. Figure 11 demonstrates that, besides skyrocketing Google search queries for the term *Bitcoin*, from 2017 onward, a sharp increase in search queries for the term *Cryptocurrency* occurred. Already from 2015 onward, rising requests for the term *Blockchain* can be seen. This signals interest in Blockchain technology on its own that has been growing over the years, as well as strongly enhanced interest in alternatives to Bitcoin from 2017 onward.

As a response to investors' growing demand for alternative investments in the cryptocurrency market, the emergence of a multitude of new digital coins ensued. Throughout 2017, the cryptocurrency market changed its structure from being dominated by Bitcoin to a more diversified market offering numerous technologies and variants of cryptocurrencies [79].

Figure 12 shows the dominance of Bitcoin at the beginning of 2017 over the complete cryptocurrency market with its market share, measured with respect to the total market capitalization of the top 1000 cryptocurrencies, being as high as 90%. Within three months after February 2017, its relative market share dropped by 50%, while its total capitalization value and that of other crypto-currencies continued to grow at accelerating pace until December 2017.

During the last two months of 2017, the overall capitalization of the crypto-market multiplied by a factor four. As the idiom claims, 'the rising tide lifts all boats', and thus, the large inflow of fresh money to the cryptocurrency market impacted most of the Altcoins as well as the still dominating Bitcoin. However, with the crash following 18 December 2017, the value of Bitcoin and that of many cryptocurrencies has been dropping, with Bitcoin losing 60% of its total value (as of February 2018), putting the Bitcoin market share to an all-time low.

One can say that 2017 was the year of the cryptocurrencies. New competitors were constantly pushing their coins as well as new tokens into the market. These coins and tokens were associated with new proposed ideas of the application of Blockchain technology. As the market is becoming ever-more complex every day, the possibility for the release of a new coin making Bitcoin obsolete is perceived more likely. By contrast, others note that Bitcoin may survive as a kind of relatively scarce digital 'commodity' or asset, used by investors as a storage of value and diversification vehicle.

We now turn to a more quantitative analysis of the various bubbles from 2012 to 2018 to recount how well these bubbles (could) have been diagnosed in real time with metrics derived from the LPPLS model presented in §2.2.

# 4. Real-time bubble identification

## 4.1. LPPLS multiscale confidence indicators as bubble diagnostics

Following the methodology of Sornette *et al.* [24] and Zhang *et al.* [25], we use the *LPPLS Confidence Indicator* as a diagnostic tool for the recognition of bubbles. In a nutshell, the LPPLS Confidence

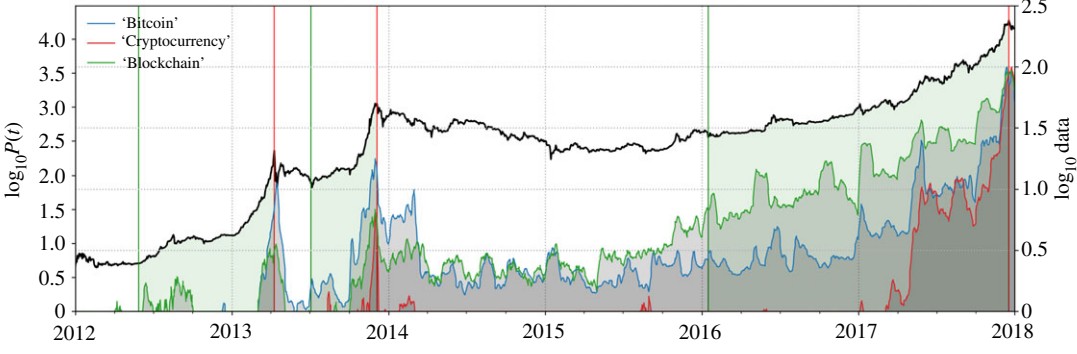

**Figure 11.** Google Trends search queries related to Bitcoin. The three grey shaded areas depict the development of search requests for the terms *Bitcoin*, *Cryptocurrency* and *Blockchain*. For studies demonstrating the relationship between such search queries and the Bitcoin price trajectory, we refer the reader to [11,12]. (Data Source: [78].)

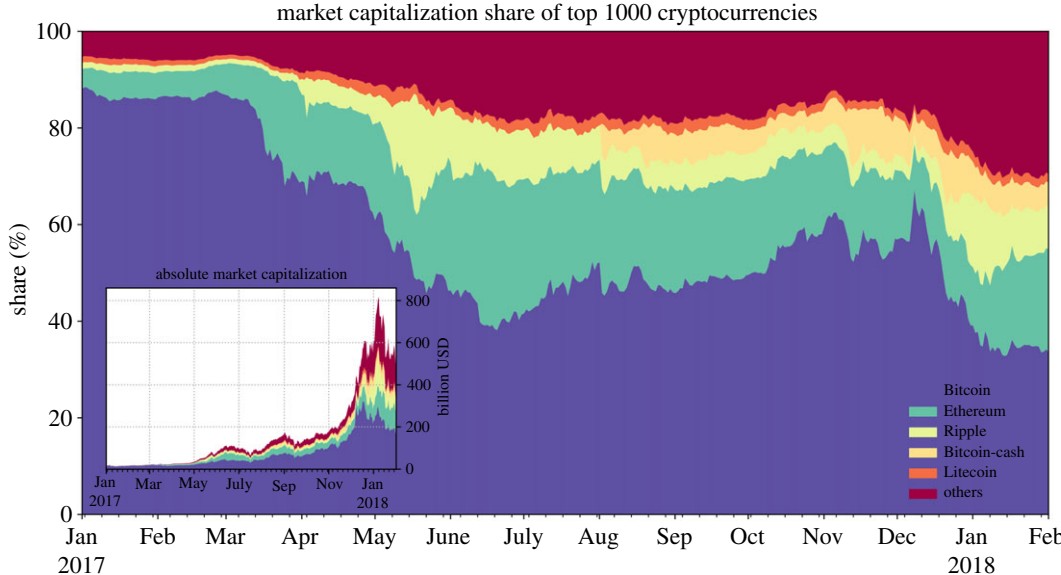

**Figure 12.** Progressive maturation of the cryptocurrency market during the year of 2017. Shown is the market share of the five largest capitalized cryptocurrencies *Bitcoin*, *Ethereum*, *Ripple*, *Bitcoin Cash* and *Litecoin*, as well as the aggregate capitalization of the top 1000 cryptocurrencies excluding the five largest ones in the year 2017 (shown as 'others'). Similarly, the inset plot depicts the evolution of the absolute market capitalization. We can see that, within 1 year, Bitcoin lost a majority of its share in the total market capitalization. This signals its possibly decreasing significance and, at the same time, the growing number of competitive cryptocurrencies in the market. (Data Source: [80].)

Indicator at a given time $t_2$ is the fraction of windows $[t_1, t_2]$, obtained by scanning $t_2 - t_1$ over a certain window size interval, for which the calibration of the log-price of Bitcoin by the LPPLS formula (2.1) passes the criteria shown in table 2. A large LPPLS Confidence Indicator value indicates that the LPPLS patterns with model parameters passing the filtering conditions are found in a large fraction of time scales at this particular time $t_2$. This is then translated into a bubble diagnostic analysis, based on the hypothesis that the LPPLS pattern is a characteristic feature of bubbles.

In order to capture the main relevant time scales of investment horizons, we propose to partition the window sizes $dt := t_2 - t_1 + 1$ over which the LPPLS Confidence Indicator is calculated into three classes: $dt \in [30, 120]$ (short time scale), $dt \in [100, 240]$ (medium time scale) and $dt \in [200, 720]$ (long time scale). The short time scale goes approximately from one month to five months. The medium time scale ranges from six months to a year (when there are approximately 250 trading days in a calendar year). The large time scale goes from 10 months to approximately 3 years. Thus, for each $t_2$, we construct three LPPLS Confidence Indicators, one for each set of window sizes. There are respectively 91, 141 and 521 differently sized windows for the short, intermediate and long time scales. We refer to the ensemble of the three indicators calculated in this way as *LPPL Multiscale Confidence Indicators*.

**Table 2.** Filter conditions for qualified LPPLS fits. The table gives the parameter bounds that were used to filter for qualified LPPL fits. The constraint on $B$ ensures the existence of a positive bubble. For $m$ and $t_c$, the boundaries are excluded to avoid singular behaviours in the search algorithm. The damping factor quantifies the allowed downwards movements of bubble fits. The $O$-parameter measures the number of oscillations that occur within the fit window $[t_1, t_2]$. The filter on $O$ is applied only when the amplitude of the oscillations, as quantified by $C$, is sufficiently large relative to the power-law amplitude $B$.

| parameter | | filter bounds | description | pre-condition |
|---|---|---|---|---|
| power-law amplitude | $B$ | $(-\infty, 0)$ | — | — |
| power-law exponent | $m$ | $(0, 1)$ | — | — |
| log-periodic frequency | $\omega$ | $[4, 25]$ | — | — |
| critical time | $t_c$ | $(0, dt_i)$ | — | — |
| damping | $D$ | $[0.5, \infty)$ | $D = \frac{m|B|}{\omega|C|}$ | — |
| number of oscillations | $O$ | $[2.5, \infty)$ | $O = \frac{\omega}{2\pi} \ln \frac{t_c - t_1}{t_c - t_2}$ | if $|C/B| \geq 0.05$ |

Figure 13 depicts the LPPLS Multiscale Confidence Indicators, as described above, constructed on the *btc/usd* price. One can observe that the short and intermediate indicators peak at or close to the corresponding bubble tops, providing useful diagnostics, in particular, for the first three pre-2014 bubbles. For the shortest bubble episodes such as Short Bubbles 5 and 6, there are no signals. This can be understood from the choice of time scale: even the short time scale $dt \in [30,120]$ is too long, compared with the duration of these two bubbles. This shows the importance of having a suitable set of indicators scanning the time scales of interest. Although not all of the minor short bubbles are identified due to the mentioned limitation of time scales, in all cases for which the LPPLS Confidence Indicator reaches its maximum value 1, a crash follows.

After 2016, the LPPLS Confidence Indicator at the long time scale exhibits an increasing amplitude, associated with the development of the third long bubble. Close to the peak in December 2017, the large-scale LPPLS Confidence Indicator is close to its maximum level, providing a warning for an imminent massive crash, following bubble price dynamics that evolved over the long term. Since the end of 2016, we have been implementing this and other indicators to watch the development of asset price bubbles, among them the many bubbles in the Bitcoin price: for instance, in our monthly report of the *Financial Crisis Observatory* on the 1 December 2017, we pointed out the alarm generated in advance of the burst of the third long bubble (see 1 December 2017 Synthesis report at www.er.ethz.ch/financial-crisis-observatory.html).

Overall, the LPPLS Multiscale Confidence Indicators over the three monitored scales provide a valuable risk management metric for early recognition of emerging bubbles at short and long time scales as well as their approaching bursts. In particular, the long bubbles are clearly identified in advance by the fact that the three LPPLS Confidence Indicators peaked jointly for these long bubbles. The other non-synchronized indicator peaks testify to the multiscale nature of bubbles in Bitcoin.

## 4.2. Predictions of bubble burst times using ensemble forecasting

### 4.2.1. Methodology

We now present systematic mock-up forecasts of the burst times of the three long and eight short bubbles identified in figure 2 and table 1. For each bubble, we position the mock-up 'present' time of analysis $t_2$ 10 business days prior to the corresponding identified bubble peak date given in table 1. Next, for each analysis time $t_2$, we identify the associated bubble start time $t_1^*$ by means of the Lagrange Regularization Methodology, as explained in §2.1 and in appendix D.

Based on the complete 'frame' of each bubble $[t_1^*, t_2]$, we perform LPPLS calibrations over all windows by scanning $t_1$ from $t_1^*$ to $t_2 - 29$. Out of these $t_2 - t_1^* - 28$ LPPLS fits, we again only keep those that pass the filter conditions listed in table 2. For a given $t_2$, each of the remaining qualified fits is then characterized by (i) its window size $dt := t_2 - t_1 + 1$ corresponding to the observed time scale and (ii) a forecasted horizon $t_c - t_2$, which is the estimated remaining time from the present time $t_2$ up to the predicted end $t_c$ of the bubble.

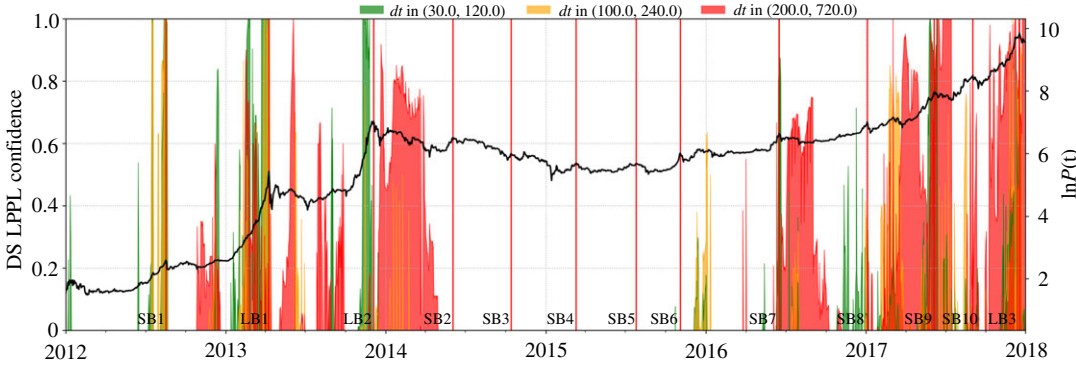

**Figure 13.** Diagnosing bubbles over multiple scales using the *LPPLS Confidence Indicator*. The LPPLS Confidence Indicator diagnoses bubbles nucleating over short (green), intermediate (yellow) and long time scales (red). The larger the value of the metric at a given point in time, the larger the portion of qualified LPPLS fits, indicating stronger bubble activity. The peaks of the bubbles, as documented in figure 2 and table 1, are depicted in red vertical lines.

Because analyses performed over different window sizes may result in different forecasts, corresponding to multiple price dynamics, we use the k-mean clustering method to identify clusters in the ($dt$, $t_c - t_2$) space whose components yield common predictions for the critical time $t_c$. Then, if a unique, well-defined cluster is identified, this can be interpreted as a consensus forecast obtained over all the time windows included in the cluster. If there are two well-defined clusters, similarly, one should conclude that the analysis suggests two different scenarios for the future development of the bubble, and so on. We augment the k-mean clustering method by using the Silhouette metric to determine the optimal number $\hat{k}$ of clusters and thus of forecasted scenarios. The details of the k-mean clustering implementation and application of the Silhouette metric are given in appendix E.

### 4.2.2. Long bubbles

Table 3 shows that, for the long bubbles, between 40 and 68% of the fits are qualified according to the filtering constraints from table 2. For the first and second long bubbles, two main scenarios are identified while for the third long bubble, we find up to nine scenarios, most of them being far to the future and with few elements, thus unreliable and discarded. Figure 14 contains three panels, one for each of the three long bubbles. Each panel shows the log-price of Bitcoin over the time interval of the full development of the bubble together with, for each of the identified clusters, the 15 LPPLS fits (when they exist) having the lowest sum of squared errors (SSE).

Figure 14*a* shows the first long bubble that peaked on 9 April 2013. The scenario corresponding to the first largest cluster predicts the mean of $t_c$ to be 14 days in the future (after $t_2$) with a standard deviation of 9 days. The second, smaller cluster predicts $t_c$ to be 262 days in the future with a standard deviation of 34 days. Hence, the first scenario is essentially telling us that the bubble peak is imminent, with a quite precise bracketing of the true peak. Interpreting the fraction of fits belonging to this cluster as a proxy for the probability for this scenario, we can attribute a probability of 80% (101 fits) for this scenario.

The second less probable (20% probability, 25 fits) scenario considers as possible that the bubble may continue much longer, up to another year, before bursting. Given the interplay between the endogenous herding dynamics captured by these scenarios and the exogenous shocks that continuously punctuated the bubble development, in retrospect, these two scenarios appear reasonable possibilities when placing oneself at the time $t_2$ of the analysis. In hindsight, the highest probability scenario was the one that unfolded.

Turning to the second long bubble that peaked on 4 December 2013 and is shown in figure 14*b*, we again identify two clusters. The first cluster predicts $t_c$ to be 4 days in the future (after $t_2$) with a standard deviation of 2 days and probability 82% (41 fits). The second cluster predicts $t_c$ to be 76 days in the future with a standard deviation of 17 days, with a probability of 18% (9 fits). In this case, the bubble peaked 10 days after $t_2$, i.e. later than predicted by the first cluster, but this discrepancy can be reconciled when noticing the double peak structure and flatness of this bubble end.

The third longest bubble that ended in December 2017 has nine clusters, but only the two most probable scenarios in terms of the number of fits contained in the associated clusters are depicted in figure 14*c*. The first cluster predicts $t_c$ to be at the time $t_2$ of analysis with all the 90 qualified fits

**Table 3.** Summary of cluster data for long and short bubbles. The first column enumerates the three long bubbles and the 10 additional peaks, which include the eight short bubbles, as qualified by the method of §2.1. The second column gives the 'present' time $t_2$ that was chosen ten business days prior to the peak of each bubble. The third column gives the percentage of fits that are qualified. The fourth column gives the optimal number of clusters. Finally, the fifth column lists the values of the Silhouette coefficient corresponding to the optimal cluster configuration. For further information on the details of the calculation, see appendix E.

| no. | analysis date $t_2$ | qualified fits (%) | clusters $\hat{k}$ | Silhouette $\bar{s}_{\hat{k}}$ |
|---|---|---|---|---|
| long bubble data | | | | |
| 1 | 28 Mar 2013 | 43 | 2 | 0.85 |
| 2 | 22 Nov 2013 | 40 | 2 | 0.84 |
| 3 | 6 Dec 2017 | 68 | 9 | 0.67 |
| short bubble data | | | | |
| 1 | 6 Aug 2012 | 59 | 2 | 0.58 |
| 2 | 22 May 2014 | 0 | — | — |
| 3 | 2 Oct 2014 | 0 | — | — |
| 4 | 27 Feb 2015 | 0 | — | — |
| 5 | 15 July 2015 | 0 | — | — |
| 6 | 23 Oct 2015 | 0 | — | — |
| 7 | 6 June 2016 | 12 | 2 | 0.91 |
| 8 | 23 Dec 2016 | 33 | 3 | 0.75 |
| 9 | 25 May 2017 | 93 | 2 | 0.57 |
| 10 | 22 Aug 2017 | 0 | — | — |

yielding the same critical time $t_c$, hence a vanishing standard deviation for the predicted $t_c$s. This scenario has a probability of 20%. The second cluster predicts $t_c$ to be 113 days in the future with a standard deviation of 7 days, with a probability of 15% (69 fits). The lower quality of these predictions resonates with their much lower probabilities, compared to the two first bubbles.

### 4.2.3. Short bubbles

Table 3 shows that, for the short bubbles, between 12 and 93% of the fits are qualified for the bubbles numbered 1 and 7–9, according to the filtering conditions listed in table 2. However, the Short Bubbles 2–6 and 10 do not have any qualified fits. This is in agreement with the previously calculated Multiscale Confidence Indicators, which show low or zero amplitude in advance of these bubbles, due to their short timespan. For the bubbles numbered 1 and 7–9, an optimal cluster configuration of two to three clusters are obtained according to the Silhouette procedure, described in appendix E.

Figure 15 contains four panels, one for each of the four short bubbles having a non-zero percentage of qualified fits. This figure is constructed in the same way as figure 14, with the condition that clusters are drawn in the plots only when they contain more than five fits.

For the Short Bubble 1 that peaked on 16 August 2012 and is shown in figure 15a, there are two clusters. The first cluster predicts $t_c$ to be 27 days in the future (after $t_2$) with a standard deviation of 2 days, with probability 51% (22 fits). The second cluster predicts $t_c$ to be 26 days in the future with a standard deviation of 2 days, with a probability of 49% (21 fits).

Next, for the Short Bubble 7 that peaked on 16 June 2016 and is shown in figure 15b, there are again two clusters, but only the first and largest one is represented as the second one has only three fits. The first cluster predicts $t_c$ to be 1 day after $t_2$ with a standard deviation of 0 days, with probability 81% (13 fits). The small value of $\sigma_{t_c}$ means that the fits in all time windows give the same $t_c$ within one day of precision.

For the Short Bubble 8 that peaked on 4 January 2017 and is shown in figure 15c, there are three clusters. The first cluster predicts $t_c$ to be 114 days after $t_2$ with a standard deviation of 9 days, with probability 51% (22 fits). The second cluster predicts $t_c$ to be 52 days in the future with a standard deviation of 10 days, with a probability of 40% (17 fits). This poor result comes from the sudden acceleration of the price during the last

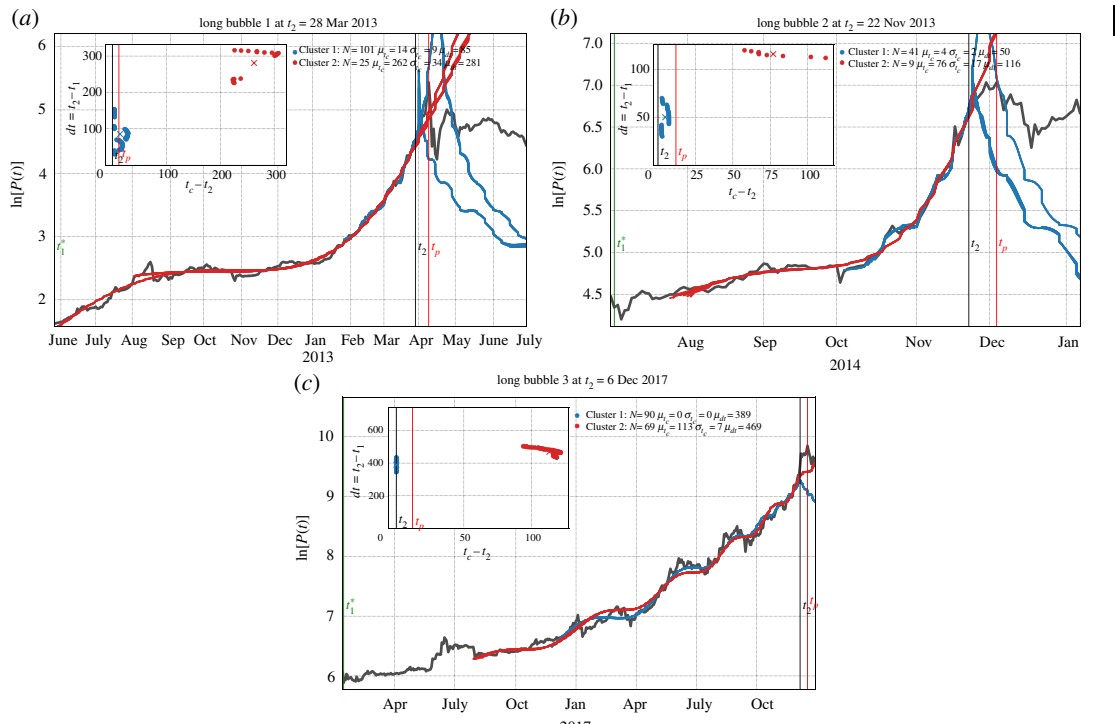

**Figure 14.** Scenario projections for the three long bubbles. Together with the log-price of Bitcoin over the time interval including the full development of each bubble, two sets of 15 LPPLS fits with formula (2.1) having the lowest sums of square errors (SSE) are shown for each of the identified clusters (blue for Cluster 1 and red for Cluster 2). The 'present' time $t_2$ of analysis is shown by the black vertical line, while the time $t_p$ at which the bubble peaked is shown by a red vertical line. $N$ represents the number of qualified fits, according to the filtering conditions in table 2 associated with each cluster. For information about the construction of the clusters, we refer to appendix E. In the legend, the pair $(\mu_{t_c}, \mu_{dt})$ represents the centroid value for the predicted $t_c - t_2$ and window sizes of Clusters 1 and 2. The corresponding standard deviations $\sigma_{t_c}$ and $\sigma_{dt}$ for the predicted $t_c$ and associated windows of each cluster are also given. The inset plots show the two largest clusters in the ($dt := t_2 - t_1 + 1$, $t_c - t_2$) space. Thereby, $t_2$ is placed at the origin on the horizontal axis and the vertical red line for $t_p$ is placed relative to $t_2$. The centroid of each cluster is indicated by a cross with the corresponding colour (blue for Cluster 1 and red for Cluster 2).

15 days before the peak, of which only the beginning phase was present in the windows of analysis. Before that sudden acceleration, the trend was informing the LPPLS calibration of a longer bubble duration. This poor result illustrates that a full operational prediction method should incorporate more time scales in order to be more reactive to such burgeoning acceleration. Again, it demonstrates the need to resolve also smaller time scales for complete short bubble detection, which may, however, become problematic when using data with a daily frequency.

For the Short Bubble 9 that peaked on 6 June 2017 and is shown in figure 15$d$, there are once more two clusters. The first cluster predicts $t_c$ to be 10 days after $t_2$ with a standard deviation of 1 day, with probability 50% (21 fits). The second cluster predicts $t_c$ to be also 10 days in the future with a standard deviation of 4 days and a probability of 50% (21 fits). This is a prediction very close to the true bubble peak.

# 5. Concluding remarks

We have focused on classifying and characterizing the history of bubbles that have developed in Bitcoin, the archetypal cryptocurrency paving the way for novel markets. We were particularly interested in Bitcoin for several reasons. First, and most importantly, in terms of market capitalization and trading volume, Bitcoin has been and is still the largest cryptocurrency, being most attractive to private and institutional investors. In the past, Bitcoin and the general cryptocurrency market have successively been opened up to a more widespread community of investors, firstly by a new generation of digital crypto-exchanges, later on through further actions undertaken by more publicly recognized exchanges such as CME Group and CBOE, both of which launched Bitcoin futures trading in late 2017. Besides its dominant prominence among cryptocurrencies, we are furthermore particularly interested in Bitcoin, because most other

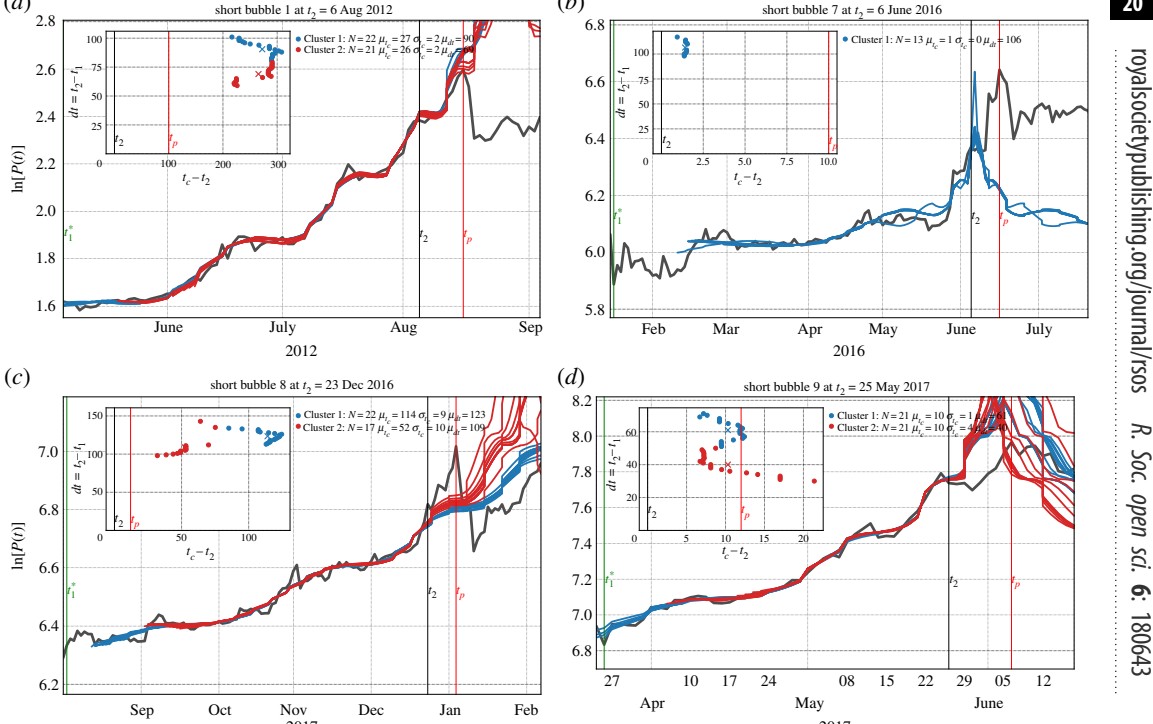

**Figure 15.** (*a–d*) Scenario projections for the four short bubbles. Same presentation as in figure 14.

cryptocurrencies have their prices highly correlated with Bitcoin's price. With its—although decreasing—enormous market share, Bitcoin seems to largely influence the overall cryptocurrency market. Thus, the statistical properties of Bitcoin are likely to be informative for other coins. In addition, Bitcoin has been among the first cryptocurrencies to be publicly traded on exchanges, giving longer available historical time series compared with other digital currencies that just started a few years ago. For these reasons and as the pioneer of a novel market, Bitcoin has been most extensively studied in the literature, so far. With this present work, we have aimed at complementing the literature on the nature and quantitative properties of the bubbles and drawdowns that have developed since 2012.

We have thus presented a detailed analysis of the dynamics of the price of Bitcoin expressed in US Dollars from January 2012 to February 2018. Given the impressive ascent of Bitcoin over this period, it is reasonable to ask whether bubbles have occurred, to characterize them if they exist and investigate the potential for their predictability.

We have been able to identify three major bubble peaks and 10 additional smaller bubble peaks that have punctuated the dynamics of Bitcoin price in the analysed time period, as well as their associated bubble start times. The statistics of the found periods of abnormal price growth suggest that the identified peaks can indeed be attributed to genuine bubbles.

Furthermore, we have presented a number of quantitative metrics and graphs to understand at least a part of the socio-economic drivers behind the evolution of Bitcoin over the period of analysis, focusing primarily on the three long bubbles. We stress in particular the attraction to Bitcoin as a novel asset type that was promoted and understood by many as a decentralized, independent medium of exchange and storage of value at the time of heavy interventions of central banks and growing scepticism about the reliability of the standard banking system. We highlight also the influence of mining and of miners' technical sophistication on Bitcoin price. The role of increasing demand from China, significantly contributing to the formation of the third long bubble, is emphasized.

Given the list of bubbles and a general understanding of their background, we proceeded by presenting a detailed quantitative, predictive analysis of these bubbles using two metrics derived from the LPPLS model.

The first one is based on the LPPLS Multiscale Confidence Indicators, defined as the fraction of qualified fits of the LPPLS model over multiple time windows. The time evolution of the LPPLS Confidence Indicators at three time scales has provided a global insight of the growing risks during bubble development.

The second approach uses a clustering method to group LPPLS fits over different time scales and predicted critical times $t_c$ (the most probable time for the start of the crash ending the bubble).

Overall, our predictive scheme seems significantly informative and useful for the three long bubbles. For three out of four short bubbles, the predictions are useful to warn of an imminent crash.

Through the described metrics, the LPPLS model has allowed us to quantitatively study Bitcoin's bubbles and their predictability. As seen, the results generated by this econophysics model are paralleled by important socio-economic events. We interpret this agreement of quantitative and qualitative analysis results as confirming evidence for the quality of the new techniques that were developed and tested. Thereby, technically, this academic study has gone one step closer to building a set of analytical tools and a testing environment towards the real-time build-up of an early warning system of bubbles.

In summary, from the quantitative bubble analysis we conclude that Bitcoin price, although highly volatile and frequently dropping sharply, has had the potential to recover from drawdowns in no time, enabling the price to grow to ever higher levels again and again during the past years. This leaves open the question whether the most recent change of regime has introduced a new era to Bitcoin's market behaviour or whether the digital currency will continue its accelerating growth to new heights in the future. We hope to report further on developments going on in the broader universe of cryptocurrencies in the future.

Data accessibility. All data used are openly available for download on the webpages of the relevant sources mentioned in the text and stated in the references section.
Authors' contributions. All authors contributed to the design, analysis and writing, with authorship according to relative contribution.
Competing interests. We declare we have no competing interests.
Funding. We received no funding for this study.

# Appendix A. The $\varepsilon$-drawup/$\varepsilon$-drawdown methodology

We start out by calculating the daily discrete log-returns

$$r_i = \ln P[t_i] - \ln P[t_{i-1}], \quad i = 1, 2, \ldots \tag{A 1}$$

with $t_i = t_0 + i\Delta_t$ and $\Delta_t = 1$ day.

The first date $t_{i_0}$ in our timeframe, corresponding to the discrete time $i_0 = 1$, is defined as the beginning of a drawup (drawdown) if $r_1 > 0$ ($r_1 < 0$). Then, for each subsequent $t_i > t_{i_0}$, we calculate the cumulative return up to this $t_i$ as

$$p_{i_0,i} = \sum_{k=i_0}^{i} r_k = \ln P[t_i] - \ln P[t_{i_0}]. \tag{A 2}$$

At each time, we need to check whether the current drawup (drawdown) phase is still active. We test this by calculating the largest deviation $\delta_{i_0,i}$ of the price trajectory from a previous maximum (minimum).

$$\delta_{i_0,i} = \begin{cases} \max_{i_0 \le k \le i} \{p_{i_0,k}\} - p_{i_0,i} & \text{for drawups} \\ p_{i_0,i} - \min_{i_0 \le k \le i} \{p_{i_0,k}\} & \text{for drawdowns.} \end{cases} \tag{A 3}$$

The procedure is stopped at time $i$ when the deviation exceeds a predefined tolerance $\varepsilon$.

$$\delta_{i_0,i} > \varepsilon. \tag{A 4}$$

The stopping tolerance quantifies how much the price is allowed to move in the direction opposite to the drawup/drawdown trend.

When the procedure has been stopped, the end of the current drawup (drawdown) phase is determined as the time of the highest (lowest) price seen in the tested interval:

$$i_1 = \begin{cases} \arg\max_{i_0 \le k \le i} \{p_{i_0,k}\} & \text{for drawups} \\ \arg\min_{i_0 \le k \le i} \{p_{i_0,k}\} & \text{for drawdowns.} \end{cases} \tag{A 5}$$

The Epsilon drawup/drawdown procedure is restarted at time $i_1 + 1$. The start of the next drawup/ drawdown period will then be located at this time. By construction of $\delta$ in equation (A 3) and the

stopping condition, a drawup (respectively, drawdown) is always followed by a drawdown (respectively, drawup). The procedure is repeated until the full length of the analysed time series is represented as a sequence of drawups and drawdowns. From the sequence identified by the Epsilon drawup/drawdown procedure, a set of peak times $\{t_{p,0}, t_{p,1}, \ldots\}$, defined as the set containing the end times of all drawup periods in the sequence, is obtained. These times can be regarded as the peaks of candidate bubbles.

A reasonable way to find suitable values for the stopping tolerance $\varepsilon$ is to incorporate the dynamics of realized return volatility and define

$$\varepsilon(\varepsilon_0, w) = \varepsilon_0 \sigma(w) \tag{A 6}$$

with realized volatility $\sigma(w)$ estimated over a moving window of the past $w$ days and $\varepsilon_0$ being a constant multiplier.

The partitioning of the drawup–drawdown sequence is strongly dependent on the choice of the pair $(\varepsilon_0, w)$. Selections resulting in large values of the tolerance $\varepsilon$ will tend to yield a coarse long-term sequence while small $\varepsilon$-values result in more frequent interruption of a drawup or drawdown, yielding finer sequences. In order to account for this dependence, instead of selecting a single pair of values, we execute the Epsilon drawup/drawdown procedure for many different pairs $\{(\varepsilon_0, w)_j | j = 1, \ldots, N_\varepsilon\}$. More specifically, we scan $\varepsilon_0$ in $[0.1, 5.0]$ in steps of $\Delta\varepsilon_0 = 0.1$ and $w$ in $[10, 60]$ in steps of $\Delta w = 5$ days. This amounts to a total number of $N_\varepsilon = 50 \cdot 11 = 550$ pairs $(\varepsilon_0, w)$.

For each pair $(\varepsilon_0, w)_j$, we obtain an associated set of peak times

$$\mathcal{T}_j = \{t_{p,0}, t_{p,1}, \ldots\}_j, \quad j = 1, \ldots, N_\varepsilon, \tag{A 7}$$

which may or may not overlap with the sets of other pairs. The collection of all peak times that have been identified at least once across all pairs is then the union of the single sets $\mathcal{T}_j$:

$$\mathcal{T} = \bigcup_{j=1}^{N_\varepsilon} \mathcal{T}_j. \tag{A 8}$$

We define $N_{t_p}$ as the number of unique peak times, i.e. the cardinal of the union set $\mathcal{T}$. Next, for each of the $N_{t_p}$ peak times in $\mathcal{T}$, we count the number of times $N_{t_{p,k}}$ that they occurred over all trials

$$N_{t_{p,k}} = \sum_{j=1}^{N_\varepsilon} I_j(t_{p,k}), \tag{A 9}$$

where $I_j$ is given by the indicator function

$$I_j(t_{p,k}) = \begin{cases} 1 & \text{if } t_{p,k} \text{ in } \mathcal{T}_j \\ 0 & \text{else.} \end{cases} \tag{A 10}$$

The index $k$ in $t_{p,k}$ registers the $k$th unique peak time in $\mathcal{T}$. We further divide the number of counts $N_{t_{p,k}}$ by the fixed total number of tested pairs $N_\varepsilon$ to obtain a series of values between $[0, 1]$, each indicating the fraction of occurrence of the corresponding peak time with respect to the total number of trials. We group these in the set

$$\mathcal{N} = \left\{ n_{t_{p,k}} = \frac{N_{t_{p,k}}}{N_\varepsilon} | k = 1, \ldots, N_{t_p} \right\}. \tag{A 11}$$

In this way, the information covered by $N_\varepsilon$ pairs is condensed down into a single figure for each peak time.

Based on the series $\mathcal{N}$, we can now search for potential bubble peak times. We introduce the conditions $0.95 \leq n_{t_{p,k}} \leq 1$ for long bubbles and $0.65 \leq n_{t_{p,k}} < 0.95$ for short bubbles. We filter for peak times that fulfil these conditions, in order to classify them as long and short bubble peaks:

$$\begin{aligned} \mathcal{T}_{LT} &= \{t_{p,k} | 0.95 \leq n_{t_{p,k}} \leq 1 \text{ for } k = 1, \ldots, N_{t_p}\} \\ \text{and} \quad \mathcal{T}_{ST} &= \{t_{p,k} | 0.65 \leq n_{t_{p,k}} < 0.95 \text{ for } k = 1, \ldots, N_{t_p}\}. \end{aligned} \tag{A 12}$$

# Appendix B. The log-periodic power-law singularity model

In a bubble regime, the observed price trajectory of a given asset decouples from its intrinsic fundamental value [35,36]. For a given fundamental value, the JLS model [17,18] assumes that the logarithm of the observable asset price $p(t)$ follows:

$$\frac{dp}{p} = \mu(t)\,dt + \sigma(t)\,dW - \kappa dj, \tag{B 1}$$

where $\mu(t)$ is the expected return, $\sigma(t)$ is the volatility, $dW$ is the infinitesimal increment of a standard Wiener process and $dj$ represents a discontinuous jump such that $j = n$ before and $j = n + 1$ after a crash occurs (where $n$ is an integer). The parameter $\kappa$ quantifies the amplitude of a possible crash.

In the network structure that is underlying the LPPL, two types of agents are considered: the first group consists of traders with rational expectations [81], while the second one is formed by noise traders. Noise traders are susceptible to show imitation and herding behaviour as a group. Their collective behaviour may destabilize asset prices. Johansen *et al.* [18] propose that the behaviour of the agent network can be incorporated by writing the *crash hazard rate* $h(t)$ in the following form:

$$h(t) = \alpha(t_c - t)^{m-1}(1 + \beta \cos(\omega \ln(t_c - t) - \phi'), \tag{B 2}$$

where $\alpha$, $\beta$, $\omega$ and $t_c$ are parameters. Equation (B 2) tells us that the risk of a crash resulting from herding behaviour is a sum of a power-law singularity $(\alpha(t_c - t)^{m-1})$ and accelerating large-scale amplitude oscillations that are periodic in the logarithm of the time to the singularity (or critical time) $t_c$. The power-law singularity embodies the positive feedback mechanism associated with the herding behaviour of noise traders. At time $t = t_c$, the power law reaches the singularity. Seyrich & Sornette [82] have recently presented a percolation-based model providing a micro-foundation for this singular behaviour. The log-periodic oscillations represent the tension and competition between the two types of agents that tend to create deviations around the faster-than-exponential price growth as the market approaches a finite-time-singularity at $t_c$.

The no-arbitrage condition imposes that the excess return $\mu(t)$ during a bubble phase is proportional to the crash hazard rate given by equation (B 2). Indeed, setting $E[dp] = 0$, and assuming that no-crash has yet occurred ($dj = 0$), this yields $\mu = \kappa h(t)$, since $E[dj] = h(t)dt$ by definition of $h(t)$. By integration, we obtain the trajectory of the expected log-price during a bubble phase, conditional on the crash not having happened yet, as

$$E[\ln p(t)] = A + B|t_c - t|^m + C|t_c - t|^m \cos(\omega \ln|t_c - t| - \phi), \tag{B 3}$$

with $B = -\kappa\alpha/m$ and $C = -\kappa\alpha\beta/\sqrt{m^2 + \omega^2}$. Note that the formula extends the price dynamics beyond $t_c$ by replacing $t_c - t$ by $|t_c - t|$, which corresponds to assuming a symmetric behaviour of the average of the log-price around the singularity at $t_c$.

Bubble regimes are in general characterized by $0 < m < 1$ and $B < 0$. The first condition $m < 1$ writes that a singularity exists (the momentum of the expected log-price diverges at $t_c$ for $m < 1$), while $m > 0$ ensures that the price remains finite at the critical time $t_c$. The second condition $B < 0$ expresses that the price is indeed growing super-exponentially towards $t_c$ (for $0 < m < 1$).

# Appendix C. Estimation of the LPPLS model

The log-price of a given instrument can be described via

$$LPPL(\phi, t) = A + B(f) + C_1(g) + C_2(h), \tag{C 1}$$

where $\phi = \{A, B, C_1, C_2, m, \omega, t_c\}$ is a $(1 \times 7)$ vector of parameters that we would like to determine and

$$f \equiv (tc - t)^m, \tag{C 2}$$

$$g \equiv (t_c - t)^m \cos(\omega \ln(t_c - t)) \tag{C 3}$$

and

$$h \equiv (t_c - t)^m \sin(\omega \ln(tc - t)). \tag{C 4}$$

Fitting equation (C 1) to the log-price time-series amounts to search for the parameter set $\phi^*$ that yields the smallest $N$-dimensional distance between realization and theory. Mathematically, using the

$L^2$ norm, we form the following sum of squares of residuals:

$$F(t_c, m, \omega, A, B, C_1, C_2) = \sum_{i=1}^{N} [\ln [P(t_i)] - A - B(f_i) - C_1(g_i) - C_2(h_i)]^2, \quad \text{(C 5)}$$

for $i = 1, \ldots, N$. We proceed in two steps. First, slaving the linear parameters $\{A, B, C_1, C_2\}$ to the remaining nonlinear parameters $\phi = \{t_c, m, \omega\}$, yields the cost function $\chi^2(\phi)$

$$\chi^2(\phi) := F_1(t_c, m, \omega) = \min_{\{A,B,C_1,C_2\}} F(t_c, m, \omega, A, B, C_1, C_2) = F(t_c, m, \omega, \widehat{A}, \widehat{B}, \widehat{C}_1, \widehat{C}_2), \quad \text{(C 6)}$$

where the hat symbol $\widehat{\ }$ indicates estimated parameters. This is obtained by solving the optimization problem

$$\{\widehat{A}, \widehat{B}, \widehat{C}_1, \widehat{C}_2\} = \arg \min_{\{A,B,C_1,C_2\}} F(t_c, m, \omega, A, B, C_1, C_2), \quad \text{(C 7)}$$

which can be obtained analytically by solving the following matrix equations:

$$\begin{bmatrix} N & \sum f_i & \sum g_i & \sum h_i \\ \sum f_i & \sum f_i^2 & \sum f_i g_i & \sum f_i h_i \\ \sum g_i & \sum f_i g_i & \sum g_i^2 & \sum g_i h_i \\ \sum h_i & \sum f_i h_i & \sum g_i h_i & \sum h_i^2 \end{bmatrix} \begin{bmatrix} \widehat{A} \\ \widehat{B} \\ \widehat{C}_1 \\ \widehat{C}_2 \end{bmatrix} = \begin{bmatrix} \sum y_i \\ \sum y_i f_i \\ \sum y_i g_i \\ \sum y_i h_i \end{bmatrix}. \quad \text{(C 8)}$$

Second, we solve the nonlinear optimization problem involving the remaining nonlinear parameters $m, \omega, t_c$,

$$\{\widehat{t_c}, \widehat{m}, \widehat{\omega}\} = \arg \min_{\{t_c,m,\omega\}} F_1(t_c, m, \omega). \quad \text{(C 9)}$$

The model is calibrated on the data using the ordinary least squares method, providing estimations of all parameters $t_c, \omega, m, A, B, C_1, C_2$ in a given time window of analysis. For each fixed data point $t_2$ (corresponding to a fictitious 'present' up to which the data are recorded), we fit the price time series in shrinking windows $(t_1, t_2)$ of length $dt := t_2 - t_1 + 1$ decreasing from 720 trading days to 30 trading days. We shift the start date $t_1$ in steps of 1 trading day, thus giving us 691 windows to analyse for each $t_2$. In order to minimize calibration problems and address the sloppiness of the model (B 3) with respect to some of its parameters (and in particular $t_c$), we use a number of filters to select the viable solutions, which are summarized in table 4. For further information about the sloppiness of the LPPLS model, we refer to [24,83,84]. These filters derive from the empirical evidence gathered in investigations of previous bubbles [24,25,85]. Only those calibrations that meet the conditions given in table 4 are considered valid and the others are discarded.

Previous calibrations of the JLS model have further shown the value of additional constraints imposed on the nonlinear parameters in order to remove spurious calibrations (false-positive identification of bubbles) [83,86,87].

# Appendix D. The Lagrange Regularization Approach

The LPPLS model is assumed to be a valid description for the log-price trajectory only if the underlying asset is in a bubble phase. Hence, if the LPPLS is fit to time periods corresponding to phases of non-bubble price growth, spurious fit results might be the consequence. Therefore, it is important to determine the start of a bubble at first, and then apply the LPPLS for time windows with start points later than the found bubble start date. A solution for the bubble start time identification problem which itself is based on application of the LPPLS model has been introduced recently by Demos & Sornette [32], who propose the Lagrange Regularization Approach.

Formally, being located at the time $t_2$, our goal is to determine the bubble start time $t_1^*$ as the time corresponding to the 'best-fit' among a set of LPPLS fits that are computed for varying fit window start times $t_1$. Commonly, one would select the 'best-fit' among such a group of fits as the one minimizing the fit cost function $\mathcal{X}^2(t_1) := \text{SSE}(t_1)/N$, where $\text{SSE}(t_1)$ is the sum of squared errors for the fit with window start time $t_1$ and $N := t_2 - t_1$ is the fit window length:

$$t_1^* = \arg \min_{t_1} \mathcal{X}^2(t_1). \quad \text{(D 1)}$$

**Table 4.** Filter conditions for qualified LPPLS fits. The table gives the parameter bounds that were used to filter for qualified LPPL fits. The constraint on $B$ ensures the existence of a positive bubble. For $m$ and $t_c$, the boundaries are excluded to avoid singular behaviours in the search algorithm. The damping factor quantifies the allowed downwards movements of bubble fits. The $O$-parameter measures the number of oscillations that occur within the fit window $[t_1, t_2]$. The filter on $O$ is applied only when the amplitude of the oscillations, as quantified by $C$, is sufficiently large relative to the power-law amplitude $B$.

| parameter | | filter bounds | description | pre-condition |
|---|---|---|---|---|
| power-law amplitude | $B$ | $(-\infty, 0)$ | — | — |
| power-law exponent | $m$ | $(0, 1)$ | — | — |
| log-periodic frequency | $\omega$ | $[4, 25]$ | — | — |
| critical time | $t_c$ | $(0, dt_i)$ | — | — |
| damping | $D$ | $[0.5, \infty)$ | $D = \frac{m|B|}{\omega|C|}$ | — |
| number of oscillations | $O$ | $[2.5, \infty)$ | $O = \frac{\omega}{2\pi} \ln \frac{t_c - t_1}{t_c - t_2}$ | if $|C/B| \geq 0.05$ |

However, small sample sizes (i.e. small fit windows) tend to achieve smaller values of the cost function, in other words they are more likely to be selected when deciding according to the criterion above. This reflects the standard problem of over-fitting when the number of data points decreases in comparison with the number of degrees of freedom (the number of adjustable parameters). In order to circumvent this problem, Demos and Sornette introduce a simple regularization term that penalizes the cost function with the size of the fit window, yielding the *modified average SSE* $\mathcal{X}_\lambda^2(t_1) := \mathcal{X}^2(t_1) - \lambda(t_2 - t_1)$, resulting in the modified optimization problem

$$t_1^* = \arg \min_{t_1} \mathcal{X}_\lambda^2(t_1). \tag{D 2}$$

The regularization parameter $\lambda$ is determined empirically by linear regression of $\mathcal{X}^2(t_1)$ on $(t_2 - t_1)$, whereby $\lambda$ is the slope of the resulting linear function. Clearly, the magnitude of $\lambda$ measures the tendency of the model to fit short windows at a smaller average SSE compared with larger ones. If $\lambda$ was large, selection of the optimal fit based on the standard SSE/$N$ would strongly favour short windows. In the detrended cost $\mathcal{X}_\lambda^2(t_1)$, this short window bias is eliminated. This allows for a fair comparison of the fit performance over different window sizes.

Hence, at a given $t_2$, we can select the optimal bubble start time $t_1^*$ by minimizing the modified cost function over all LPPLS fits conducted at that $t_2$. With the bubble start time determined, we discard all fits that have their start before the bubble start time, i.e. all fits with $t_1 < t_1^*$. Ultimately, in order to make the estimate of $t_1^*$ robust, we can perform the procedure for groups of LPPLS fits conducted at different fixed window end times $t_2$. Ideally, we would thereby like to obtain a stable estimate for $t_1^*$ over varying $t_2$. Evidence for the rigidity of the bubble start time $t_1^*$ with respect to the time of analysis $t_2$ is indeed demonstrated in [83], which supports the usefulness of the procedure.

# Appendix E. k-means clustering and the Silhouette metric

The *k*-means algorithm aims at grouping a set of $N$ measured data points $X = \{x_1, x_2, x_3, ..., x_n\}$ into a predefined number $k \geq 2$ of clusters $\mathcal{C}_k = \{C_1, C_2, ..., C_k\}$, $C_i \subset X$, so as to minimize the variance of each cluster. The clusters are non-overlapping, $C_i \cap C_j = \emptyset$, i.e. no data point can be assigned to more than one cluster. Furthermore, any data point is contained in one of the clusters, $cup_{i=1}^k C_i = X$.

Let $\mu = \{\mu_1, \mu_2, ..., \mu_k\}$ be the centres of the clusters. They are calculated as the empirical mean of the data points in each cluster

$$\mu_i = \frac{1}{N_i} \sum_{x_j \in C_i} x_j \tag{E 1}$$

with $\{N_1, N_2, ..., N_k\}$ representing the number of members in each cluster, summing up in total to $N = \sum N_i$.

Based on the minimum variance condition, the optimal cluster configuration $\hat{\mathcal{C}}_k$ is obtained by solving the following minimization problem:

$$\hat{\mathcal{C}}_k = \arg\min_{C} \sum_{i=1}^{k} \sum_{x_j \in C_i} \|x_j - \mu_i\|^2, \tag{E 2}$$

where $\|\cdot\|$ denotes the Euclidean distance measure. Indeed, the objective function is minimized when each point is assigned to the cluster to whose centre $\mu_i$ its distance is minimal.

At the beginning, no points are assigned to any cluster. Thus, the algorithm must be initialized by randomly selecting or guessing the positions of the $k$ cluster centres. The optimal positions of the cluster centres are then iteratively calculated. In each iteration, equation (E 2) is solved. As long as this still changes the resulting composition of clusters, subsequently, the cluster centres are recalculated based on the new data point clusters according to equation (E 1). As soon as no more data points are reassigned to new clusters, the algorithm stops.

The choice of the number of clusters $k$ largely affects the capability of the results to accurately reflect the underlying structure of the data. If it is chosen too small, the variance of clusters might blow up, while an overly large number of clusters might cause unnecessarily fine data separation and excessive usage of computational resources. A tool for the 'right' selection of the number of clusters is the *Silhouette metric*.

In order to optimize the number of clusters, we firstly calculate cluster configurations for various $k$ ranging from 2 to 10. We then employ the Silhouette metric to determine the optimal number of clusters $\hat{k}$.

(1) The optimal cluster configuration $\hat{\mathcal{C}}_k$ for a fixed value of $k$ is determined.
(2) For each data point $x_i$ in the resulting $k$ clusters, its average distance $a_i$ to all other points in its own cluster is calculated.
(3) In the same manner, the average distance of each data point $x_i$ to the data points in each of the other clusters is calculated. The minimum such value with respect to all clusters, which is the critical one, is denoted by $b_i$.
(4) The Silhouette coefficient is defined as

$$s_i = \frac{b_i - a_i}{\max(a_i, b_i)} \tag{E 3}$$

for each individual data point $x_i$.
(5) From the individual Silhouette scores, the mean Silhouette score $\bar{s}_k$ of all data points is computed for the respective cluster configuration.

The Silhouette coefficient of a data point is a measure of how closely it is related to data within its own or another cluster. By construction, it is bounded between −1 and 1. We wish to find the cluster configuration with minimal dissimilarity of each data point to its own cluster. This translates into the condition that the average Silhouette score should be as close to one as possible. The optimal number of clusters is then given by

$$\hat{k} = \arg\min_{k} \{1 - \bar{s}_k\}. \tag{E 4}$$

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
