## [Reviewer comments · Royal Society Open Science]

Review History

RSOS-180643.R0 (Original submission)

Review form: Reviewer 1

Is the manuscript scientifically sound in its present form?

No

Are the interpretations and conclusions justified by the results?

No

Is the language acceptable?

Yes

Is it clear how to access all supporting data?

Yes

Do you have any ethical concerns with this paper?

No

Have you any concerns about statistical analyses in this paper?

No

Recommendation?

Major revision is needed (please make suggestions in comments)

Comments to the Author(s)

See attached file (Appendix A).

Decision letter (RSOS-180643.R0)

10-Jul-2018

Dear Mr Gerlach,

The editors assigned to your paper ("Dissection of Bitcoin's Multiscale Bubble History from January 2012 to February 2018") have now received comments from reviewers. We would like you to revise your paper in accordance with the referee and Associate Editor suggestions which can be found below (not including confidential reports to the Editor). Please note this decision does not guarantee eventual acceptance.

Please submit a copy of your revised paper before 02-Aug-2018. Please note that the revision deadline will expire at 00.00am on this date. If we do not hear from you within this time then it will be assumed that the paper has been withdrawn. In exceptional circumstances, extensions may be possible if agreed with the Editorial Office in advance. We do not allow multiple rounds of revision so we urge you to make every effort to fully address all of the comments at this stage. If deemed necessary by the Editors, your manuscript will be sent back to one or more of the original reviewers for assessment. If the original reviewers are not available, we may invite new reviewers.

If your study uses humans or animals please include details of the ethical approval received, including the name of the committee that granted approval. For human studies please also detail

whether informed consent was obtained. For field studies on animals please include details of all permissions, licences and/or approvals granted to carry out the fieldwork.

- Data accessibility

If you wish to submit your supporting data or code to Dryad (<http://datadryad.org/>), or modify your current submission to dryad, please use the following link:
<http://datadryad.org/submit?journalID=RSOS&manu=RSOS-180643>

- Competing interests

- Authors' contributions

- Acknowledgements

- Funding statement

Please note that Royal Society Open Science charge article processing charges for all new submissions that are accepted for publication. Charges will also apply to papers transferred to Royal Society Open Science from other Royal Society Publishing journals, as well as papers submitted as part of our collaboration with the Royal Society of Chemistry (<http://rsos.royalsocietypublishing.org/chemistry>). If your manuscript is newly submitted and subsequently accepted for publication, you will be asked to pay the article processing charge, unless you request a waiver and this is approved by Royal Society Publishing. You can find out

more about the charges at <http://rsos.royalsocietypublishing.org/page/charges>. Should you have any queries, please contact openscience@royalsociety.org.

on behalf of Professor Andreas Kyprianou (Associate Editor) and Prof. Mark Chaplain (Subject Editor)
openscience@royalsociety.org

Associate Editor's comments (Professor Andreas Kyprianou):

Associate Editor: 1

Comments to the Author:

We have a good report, with lots of commentary. There are a number of major issues that need ironing out before we can go to publication. Please prepare changes with an indication of how you have addressed the referee's concerns.

Comments to Author:

Reviewers' Comments to Author:

Reviewer: 1

Comments to the Author(s)

See attached file

Author's Response to Decision Letter for (RSOS-180643.R0)

See Appendix B.

RSOS-180643.R1 (Revision)

Review form: Reviewer 2

Is the manuscript scientifically sound in its present form?

Yes

Are the interpretations and conclusions justified by the results?

Yes

Is the language acceptable?

Yes

Is it clear how to access all supporting data?

Yes

Do you have any ethical concerns with this paper?

No

Have you any concerns about statistical analyses in this paper?

Yes

Recommendation?

Major revision is needed (please make suggestions in comments)

Comments to the Author(s)

Review of Bitcoin's multiscale bubble history from January 2012 to February 2018

Overall summary

This paper describes the use of the classical log-periodic power law model to analyse speculative bubbles in Bitcoin markets – itself an emerging topic of independent research interest in finance and economics. Whilst I have some concerns that more advanced methods are ultimately possible, e.g. based on extensions of the model given in Appendix B, I think it would be important to see this group publish their perspective of such an important topic in a good journal such as this.

Comments on previous revision

I felt that some of the previous reviewer's comments were a bit too harsh e.g. the application of speculative bubble models to Bitcoin and cryptocurrency data is now well-established. As is the application of econophysics models such as this. See for instance Cheah and Fry (2015), Fry and Cheah (2016) and Fry (2018). Note that these are papers published in mainstream economics and finance journals – here the publications in the economics journal are particularly noteworthy. Whilst I am easy-going about whether the authors choose to cite these in a revision the authors might find Fry and Cheah (2016) useful as a source of references both on previous econophysics models applied to Bitcoin and the early economics and finance literature on Bitcoin.

A wholesale check of the article references should be performed e.g. the reference by Fantazzini [10] is already published. Similarly, the article by Glazer et al. has been published in a journal.

Required changes

Some more details of the Lagrange regularisation approach are needed. This seems to me to be a penalised regression approach with a penalty applied to penalise against choosing a time window that is too short?

Section 2.1. More explanation is required as to the choice of ϵ_0 and w . Why were these numbers chosen?

More explanation is also needed as regards how the short-bubble/long-bubble typology depends on the stated value of f_t . It is not clear where this comes from. In principle I like this distinction

between short-bubbles and long-bubbles but I think this needs to be placed on a firmer foundation.

I like the idea behind Section 3. This might need a bit better positioning in terms of

(a) Justified as a historical study of Bitcoin the archetypal cryptocurrency and of independent financial and economic interest in its own right

(b) Similar analyses have already been undertaken before (see e.g. Fry and Cheah, 2016)

There are necessarily some subjective elements here as identified by the previous reviewer but I don't think that this is necessarily a problem.

Need to provide a reference for the comment about a Silicon Valley based investor.

I think the orientation of Section 4 is wrong and this section should be replaced with something more in keeping with the overall aims of the paper. If this paper is for the most part a historical analysis of Bitcoin then I think this issue of real-time analysis is slightly tangential and out of place. What I think would be more useful in its place is to include an additional robustness check of the long and short bubbles identified in Section 3 using a more standard econometric methodology. Possibly something as simple as a standard event study. I was a little concerned about how the referee comment about using alternative methods was seemingly casually dismissed in the original rebuttal. The method of Bai and Perron is implemented in an R package, *Strucchange* I think, and might be used to test for a change in the mean or the variance of the log-returns. However, I suspect a better approach here would be to use a simple event study methodology applied to the start-date of each of the identified short and long bubbles.

The conclusions section should be shortened and better focussed to the contribution of the paper - namely a retrospective analysis and typology of historical bitcoin bubbles by somebody whose work I've always admired.

Do check the references as I think there are probably multiple mistakes here. (I realise this may be harder to get right than in other papers due to the abundance of electronic-only references and the fast-paced nature of the subject).

References

Cheah, E-T. and Fry, J. M. (2015) Speculative bubbles in Bitcoin markets? An empirical investigation into the fundamental value of Bitcoin. *Economics Letters* 130 32-36

Fry, J. (2018) Booms, busts and heavy-tails: the story of Bitcoin and cryptocurrency markets? *Economics Letters* 171 225-229.

Fry, J. and Cheah, E-T. (2016) Negative bubbles and shocks in cryptocurrency markets. *International Review of Financial Analysis* 47 343-352.

Decision letter (RSOS-180643.R1)

25-Jan-2019

Dear Mr Gerlach:

Manuscript ID RSOS-180643.R1 entitled "Dissection of Bitcoin's Multiscale Bubble History from January 2012 to February 2018" which you submitted to Royal Society Open Science, has been reviewed. The comments of the reviewer(s) are included at the bottom of this letter.

Please submit a copy of your revised paper before 17-Feb-2019. Please note that the revision deadline will expire at 00.00am on this date. If we do not hear from you within this time then it will be assumed that the paper has been withdrawn. In exceptional circumstances, extensions may be possible if agreed with the Editorial Office in advance.

As you will be aware, the journal does not allow multiple rounds of revision; however, on this occasion, the Editors consider you have made 'good faith' efforts to respond to the previous round of criticisms. No further revisions will be possible, so we urge you to make every effort to fully address all of the comments at this stage. If deemed necessary by the Editors, your manuscript will be sent back to one or more of the original reviewers for assessment. If the original reviewers are not available we may invite new reviewers.

- Ethics statement

- Data accessibility

- Competing interests

- Authors' contributions

All submissions, other than those with a single author, must include an Authors' Contributions section which individually lists the specific contribution of each author. The list of Authors should meet all of the following criteria; 1) substantial contributions to conception and design, or

acquisition of data, or analysis and interpretation of data; 2) drafting the article or revising it critically for important intellectual content; and 3) final approval of the version to be published.

- Acknowledgements

- Funding statement

Kind regards,
Andrew Dunn
Senior Publishing Editor
Royal Society Open Science
openscience@royalsociety.org

on behalf of Professor Andreas Kyprianou (Associate Editor) and Mark Chaplain (Subject Editor)
openscience@royalsociety.org

Associate Editor Comments to Author (Professor Andreas Kyprianou):

From the referee:

Review of Bitcoin's multiscale bubble history from January 2012 to February 2018

Overall summary

This paper describes the use of the classical log-periodic power law model to analyse speculative bubbles in Bitcoin markets – itself an emerging topic of independent research interest in finance and economics. Whilst I have some concerns that more advanced methods are ultimately possible, e.g. based on extensions of the model given in Appendix B, I think it would be important to see this group publish their perspective of such an important topic in a good journal such as this.

Comments on previous revision

I felt that some of the previous reviewer's comments were a bit too harsh e.g. the application of speculative bubble models to Bitcoin and cryptocurrency data is now well-established. As is the

application of econophysics models such as this. See for instance Cheah and Fry (2015), Fry and Cheah (2016) and Fry (2018). Note that these are papers published in mainstream economics and finance journals – here the publications in the economics journal are particularly noteworthy. Whilst I am easy-going about whether the authors choose to cite these in a revision the authors might find Fry and Cheah (2016) useful as a source of references both on previous econophysics models applied to Bitcoin and the early economics and finance literature on Bitcoin.

A wholesale check of the article references should be performed e.g. the reference by Fantazzini [10] is already published. Similarly, the article by Glazer et al. has been published in a journal.

Required changes

Some more details of the Lagrange regularisation approach are needed. This seems to me to be a penalised regression approach with a penalty applied to penalise against choosing a time window that is too short?

Section 2.1. More explanation is required as to the choice of ϵ_0 and w . Why were these numbers chosen?

More explanation is also needed as regards how the short-bubble/long-bubble typology depends on the stated value of f_t . It is not clear where this comes from. In principle I like this distinction between short-bubbles and long-bubbles but I think this needs to be placed on a firmer foundation.

I like the idea behind Section 3. This might need a bit better positioning in terms of
 (a) Justified as a historical study of Bitcoin the archetypal cryptocurrency and of independent financial and economic interest in its own right
 (b) Similar analyses have already been undertaken before (see e.g. Fry and Cheah, 2016)

There are necessarily some subjective elements here as identified by the previous reviewer but I don't think that this is necessarily a problem.

Need to provide a reference for the comment about a Silicon Valley based investor.

I think the orientation of Section 4 is wrong and this section should be replaced with something more in keeping with the overall aims of the paper. If this paper is for the most part a historical analysis of Bitcoin then I think this issue of real-time analysis is slightly tangential and out of place. What I think would be more useful in its place is to include an additional robustness check of the long and short bubbles identified in Section 3 using a more standard econometric methodology. Possibly something as simple as a standard event study. I was a little concerned about how the referee comment about using alternative methods was seemingly casually dismissed in the original rebuttal. The method of Bai and Perron is implemented in an R package, *Strucchange* I think, and might be used to test for a change in the mean or the variance of the log-returns. However, I suspect a better approach here would be to use a simple event study methodology applied to the start-date of each of the identified short and long bubbles.

The conclusions section should be shortened and better focussed to the contribution of the paper – namely a retrospective analysis and typology of historical bitcoin bubbles by somebody whose work I've always admired.

Do check the references as I think there are probably multiple mistakes here. (I realise this may be harder to get right than in other papers due to the abundance of electronic-only references and the fast-paced nature of the subject).

References

- Cheah, E-T. and Fry, J. M. (2015) Speculative bubbles in Bitcoin markets? An empirical investigation into the fundamental value of Bitcoin. *Economics Letters* 130 32-36
- Fry, J. (2018) Booms, busts and heavy-tails: the story of Bitcoin and cryptocurrency markets? *Economics Letters* 171 225-229.
- Fry, J. and Cheah, E-T. (2016) Negative bubbles and shocks in cryptocurrency markets. *International Review of Financial Analysis* 47 343-352.

Reviewer comments to Author:

Reviewer: 2

Comments to the Author(s)

Review of Bitcoin's multiscale bubble history from January 2012 to February 2018

Overall summary

This paper describes the use of the classical log-periodic power law model to analyse speculative bubbles in Bitcoin markets – itself an emerging topic of independent research interest in finance and economics. Whilst I have some concerns that more advanced methods are ultimately possible, e.g. based on extensions of the model given in Appendix B, I think it would be important to see this group publish their perspective of such an important topic in a good journal such as this.

Comments on previous revision

I felt that some of the previous reviewer's comments were a bit too harsh e.g. the application of speculative bubble models to Bitcoin and cryptocurrency data is now well-established. As is the application of econophysics models such as this. See for instance Cheah and Fry (2015), Fry and Cheah (2016) and Fry (2018). Note that these are papers published in mainstream economics and finance journals – here the publications in the economics journal are particularly noteworthy. Whilst I am easy-going about whether the authors choose to cite these in a revision the authors might find Fry and Cheah (2016) useful as a source of references both on previous econophysics models applied to Bitcoin and the early economics and finance literature on Bitcoin.

A wholesale check of the article references should be performed e.g. the reference by Fantazzini [10] is already published. Similarly, the article by Glazer et al. has been published in a journal.

Required changes

Some more details of the Lagrange regularisation approach are needed. This seems to me to be a penalised regression approach with a penalty applied to penalise against choosing a time window that is too short?

Section 2.1. More explanation is required as to the choice of ϵ_0 and w . Why were these numbers chosen?

More explanation is also needed as regards how the short-bubble/long-bubble typology depends on the stated value of f_t . It is not clear where this comes from. In principle I like this distinction between short-bubbles and long-bubbles but I think this needs to be placed on a firmer foundation.

I like the idea behind Section 3. This might need a bit better positioning in terms of

- (a) Justified as a historical study of Bitcoin the archetypal cryptocurrency and of independent financial and economic interest in its own right
 (b) Similar analyses have already been undertaken before (see e.g. Fry and Cheah, 2016)

There are necessarily some subjective elements here as identified by the previous reviewer but I don't think that this is necessarily a problem.

Need to provide a reference for the comment about a Silicon Valley based investor.

I think the orientation of Section 4 is wrong and this section should be replaced with something more in keeping with the overall aims of the paper. If this paper is for the most part a historical analysis of Bitcoin then I think this issue of real-time analysis is slightly tangential and out of place. What I think would be more useful in its place is to include an additional robustness check of the long and short bubbles identified in Section 3 using a more standard econometric methodology. Possibly something as simple as a standard event study. I was a little concerned about how the referee comment about using alternative methods was seemingly casually dismissed in the original rebuttal. The method of Bai and Perron is implemented in an R package, *Strucchange* I think, and might be used to test for a change in the mean or the variance of the log-returns. However, I suspect a better approach here would be to use a simple event study methodology applied to the start-date of each of the identified short and long bubbles.

The conclusions section should be shortened and better focussed to the contribution of the paper – namely a retrospective analysis and typology of historical bitcoin bubbles by somebody whose work I've always admired.

Do check the references as I think there are probably multiple mistakes here. (I realise this may be harder to get right than in other papers due to the abundance of electronic-only references and the fast-paced nature of the subject).

References

- Cheah, E-T. and Fry, J. M. (2015) Speculative bubbles in Bitcoin markets? An empirical investigation into the fundamental value of Bitcoin. *Economics Letters* 130 32-36
 Fry, J. (2018) Booms, busts and heavy-tails: the story of Bitcoin and cryptocurrency markets? *Economics Letters* 171 225-229.
 Fry, J. and Cheah, E-T. (2016) Negative bubbles and shocks in cryptocurrency markets. *International Review of Financial Analysis* 47 343-352.

Author's Response to Decision Letter for (RSOS-180643.R1)

See Appendix C.

RSOS-180643.R2 (Revision)

Review form: Reviewer 2

Is the manuscript scientifically sound in its present form?

Yes

Are the interpretations and conclusions justified by the results?

Yes

Is the language acceptable?

Yes

Is it clear how to access all supporting data?

Yes

Do you have any ethical concerns with this paper?

No

Have you any concerns about statistical analyses in this paper?

No

Recommendation?

Accept with minor revision (please list in comments)

Comments to the Author(s)

Review of Bitcoin's multiscale bubble history from January 2012 to February 2018

Overall comments

I would be happy to recommend publication of a suitably revised manuscript. I would also like to acknowledge the efforts of the author team in revising the previous version. However, there are a small number of minor changes I would like to see first. The main thing here is that I think a couple of minor changes in writing style would be more in-keeping with a finance paper.

Required changes**Section 2.1**

This now reads much more clearly. I think it would be worth just adding a sentence on the financial interpretation of these parameters. For example, ϵ_0 accounts for anything up to a 5 standard deviation event and for a regular financial market that trades 5 days a week corresponds to intervals of between 2 weeks to 12 weeks.

Appendix E

I like the Appendix E. However, I think this would be better in the main manuscript e.g. Section 4.3 Additional robustness check. This would give your article more authority in terms of a finance article. It also seems a shame to have this work hiding at the end of an Appendix.

References

Could you check references 10, 20 and 41 and possibly the remainder. One of the things to point out here is that SSRN is an online repository rather than an electronic journal and these articles may be more properly regarded as preprints. Do check though that these articles have not been formally published in a journal.

Conclusions section

In order to bolster the finance contribution of the paper I think it would be worthwhile to clarify in the first paragraph of the conclusions section:

- From an academic perspective the study of Bitcoin as the archetypal cryptocurrency is important from an economic and a finance perspective.
- Approach also adds to econophysics analyses of the area that have been conducted before

Possible suggestion

The conclusions section is now much more clearly written. As regards the final paragraph you could consider reading Dowd (2014) where one of the concerns raised is that Bitcoin may ultimately prove vulnerable to competitors. Another issue that may be worth considering is the ultimate vulnerability of these currencies to national regulation – e.g. the new Swiss e-Franc, new cryptocurrencies backed by large investment banks intended to supersede Ripple etc.

Dowd, K. (2014). New private monies – a bit-part player? Institute for Economic Affairs, London.

Decision letter (RSOS-180643.R2)

07-May-2019

Dear Mr Gerlach:

On behalf of the Editors, I am pleased to inform you that your Manuscript RSOS-180643.R2 entitled "Dissection of Bitcoin's Multiscale Bubble History from January 2012 to February 2018" has been accepted for publication in Royal Society Open Science subject to minor revision in accordance with the referee suggestions. Please find the referees' comments at the end of this email.

The reviewers and Subject Editor have recommended publication, but also suggest some minor revisions to your manuscript. Therefore, I invite you to respond to the comments and revise your manuscript.

- Ethics statement

- Data accessibility

If you wish to submit your supporting data or code to Dryad (<http://datadryad.org/>), or modify your current submission to dryad, please use the following link:
<http://datadryad.org/submit?journalID=RSOS&manu=RSOS-180643.R2>

- Competing interests

- Authors' contributions

- Acknowledgements

- Funding statement

Because the schedule for publication is very tight, it is a condition of publication that you submit the revised version of your manuscript before 16-May-2019. Please note that the revision deadline will expire at 00.00am on this date. If you do not think you will be able to meet this date please let me know immediately.

- 1) A text file of the manuscript (tex, txt, rtf, docx or doc), references, tables (including captions) and figure captions. Do not upload a PDF as your "Main Document".
- 2) A separate electronic file of each figure (EPS or print-quality PDF preferred (either format should be produced directly from original creation package), or original software format)

- 3) Included a 100 word media summary of your paper when requested at submission. Please ensure you have entered correct contact details (email, institution and telephone) in your user account
- 4) Included the raw data to support the claims made in your paper. You can either include your data as electronic supplementary material or upload to a repository and include the relevant doi within your manuscript
- 5) All supplementary materials accompanying an accepted article will be treated as in their final form. Note that the Royal Society will neither edit nor typeset supplementary material and it will be hosted as provided. Please ensure that the supplementary material includes the paper details where possible (authors, article title, journal name).

on behalf of Professor Andreas Kyprianou (Associate Editor) and Mark Chaplain (Subject Editor)
openscience@royalsociety.org

Associate Editor Comments to Author (Professor Andreas Kyprianou):

Associate Editor: 1

Comments to the Author:

Please address the final comments of the referee

Reviewer comments to Author:

Reviewer: 2

Comments to the Author(s)

Review of Bitcoin's multiscale bubble history form January 2012 to February 2018

Overall comments

I would be happy to recommend publication of a suitably revised manuscript. I would also like to acknowledge the efforts of the author team in revising the previous version. However, there are a small number of minor changes I would like to see first. The main thing here is that I think a couple of minor changes in writing style would be more in-keeping with a finance paper.

Required changes

Section 2.1

This now reads much more clearly. I think it would be worth just adding a sentence on the financial interpretation of these parameters. For example, ϵ_0 accounts for anything up to a 5 standard deviation event and for a regular financial market that trades 5 days a week w corresponds to intervals of between 2 weeks to 12 weeks.

Appendix E

I like the Appendix E. However, I think this would be better in the main manuscript e.g. Section 4.3 Additional robustness check. This would give your article more authority in terms of a finance article. It also seems a shame to have this work hiding at the end of an Appendix.

References

Could you check references 10, 20 and 41 and possibly the remainder. One of the things to point out here is that SSRN is an online repository rather than an electronic journal and these articles may be more properly regarded as preprints. Do check though that these articles have not been formally published in a journal.

Conclusions section

In order to bolster the finance contribution of the paper I think it would be worthwhile to clarify in the first paragraph of the conclusions section:

- From an academic perspective the study of Bitcoin as the archetypal cryptocurrency is important from an economic and a finance perspective.
- Approach also adds to econophysics analyses of the area that have been conducted before

Possible suggestion

The conclusions section is now much more clearly written. As regards the final paragraph you could consider reading Dowd (2014) where one of the concerns raised is that Bitcoin may ultimately prove vulnerable to competitors. Another issue that may be worth considering is the ultimate vulnerability of these currencies to national regulation – e.g. the new Swiss e-Franc, new cryptocurrencies backed by large investment banks intended to supersede Ripple etc.

Dowd, K. (2014). New private monies – a bit-part player? Institute for Economic Affairs, London.

Author's Response to Decision Letter for (RSOS-180643.R2)

See Appendix D.

Decision letter (RSOS-180643.R3)

04-Jun-2019

Dear Mr Gerlach,

I am pleased to inform you that your manuscript entitled "Dissection of Bitcoin's Multiscale

Bubble History from January 2012 to February 2018" is now accepted for publication in Royal Society Open Science.

on behalf of Professor Andreas Kyprianou (Associate Editor) and Mark Chaplain (Subject Editor)
openscience@royalsociety.org

Appendix A

This paper addresses a topic of growing interest, which is volatility of cryptoasset prices (specifically bitcoin in this paper). While the paper is topical and has potential merit, it possesses too many problems in the form of inaccuracies, omissions, imprecision, and unsubstantiated conclusions to be suitable for publication in its present state.

I am very concerned about some of the conclusions in Section III, where the authors attempt to link certain events to changes in bitcoin's price and fall victim to the very common problem (outside of academia) of mistaking correlation is causation. For example, the authors state that "The history of Bitcoin at its early stage was highly influenced by fiscal and monetary measures undertaken during the Eurozone crisis" without providing robust empirical evidence to support this view. The authors specifically reference the Cyprus deposit haircut crisis as explaining an increase in bitcoin's price (p. 10 "Cypriot Crises that actually initiated the bubble"). However, the authors again produce no empirical evidence that buyers of bitcoin were acting on the Cyprus news, only that increased transaction activity and an increase in price coincided with events in Cyprus. In contrast, others have suggested that large buying unrelated to Cyprus in March-April 2013 by a particular Silicon Valley-based investor is what drove the price increase at this time. There is also no mention of market manipulation, only changes in fundamental demand, throughout this section as a possible explanation of the price action.

I find some of the descriptive terminology employed in the paper confusing. For example, a period of relatively low volatility / flat prices (2015) is referred to as a "normal price growth" phase. I also question the usefulness of using the term 'bubble' throughout the paper given the debate over what constitutes a bubble. For many 'bubbles' or 'manias', such as tulips or blatant frauds, we see prices collapse and never rebound. To my knowledge no other instrument has shown the type of price action as bitcoin. Perhaps the term 'correction' would be more useful, appropriate, and less controversial.

In terms of methods, the authors complain of "unpleasant arbitrariness" in the various definitions of bubbles, but then proceed to provide what appears to be arbitrary hurdles for what is a "large peak" and "intermediate peak", respectively. I would also be interested in learning how the particular method employed here for dating price regime changes compares with other methods for dating structural breaks, such as Bai and Perron, Banerjee et al, Zeileis et al; no mention of these alternative methods is made in the paper.

There are a number of inaccuracies that suggest the authors have an incomplete understanding of bitcoin and its history. For example, in the opening paragraph the authors state "Given the turbulent market history that Bitcoin has undergone since its inception in 2008", when in fact bitcoins did not have a regularly quoted price or 'market history' (meaning they did not trade actively on an exchange) until 2010. Later, it is stated inaccurately that "Chinese exchanges such as Huobi and OKCoin moving their business abroad", when both firms are still quite active (e.g., hiring additional staff) inside China since the "ban". Some claims relating to China, in addition to being inaccurate, lack novelty.

The source of the price data is not provided and needs to be provided, along with other information (e.g., frequency). Better referencing is also required (e.g., p. 3 which pundits have compared bitcoin to tulips and in what source?) There is also a disconcerting level of imprecision for a scientific paper (e.g., p. 2 “loss from the peak of about two thirds” – the actual figure is 70%).

A suggestion: some additional data in the tables would also be useful (e.g., see below example for other useful data). Also, some of the below data does not agree with the paper data, but I cannot validate as I do not know what price data source the author is using.

Correction Period	# Days	Bitcoin High	Bitcoin Low	% Decline	% Return to New High	New High Date	# Days to New High
12/17/2017 to 2/6/2018	51	\$ 19,783	\$ 5,922	-70%	?	?	?
11/8/2017 to 11/12/2017	4	\$ 7,879	\$ 5,507	-30%	43%	11/16/2017	8
9/2/2017 to 9/15/2017	13	\$ 5,014	\$ 2,951	-41%	70%	12/10/2017	40
6/11/2017 to 7/16/2017	35	\$ 3,025	\$ 1,837	-39%	65%	5/8/2017	55
3/10/2017 to 3/24/2017	14	\$ 1,326	\$ 892	-33%	49%	4/27/2017	48
11/30/2013 to 1/14/2015	410	\$ 1,166	\$ 170	-85%	585%	2/23/2017	1181
4/10/2013 to 7/7/2013	88	\$ 266	\$ 63	-76%	323%	7/11/2013	211
6/8/2011 to 11/17/2011	162	\$ 32	\$ 1.99	-94%	1504%	2/28/2013	631
5/13/2011 to 5/21/2011	8	\$ 8.45	\$ 5.58	-34%	51%	5/25/2011	12
2/10/2011 to 4/4/2011	53	\$ 1.10	\$ 0.56	-49%	96%	4/17/2011	66
11/6/2010 to 11/10/2010	4	\$ 0.50	\$ 0.14	-72%	257%	1/31/2011	86
9/14/2010 to 10/8/2010	24	\$ 0.17	\$ 0.01	-94%	1600%	10/24/2010	40
Average - Top 5 declines	138						
Median - Top 5 declines	88						

Appendix B

Dear Reviewer and Editors,

First of all, we thank you for taking the time to review our manuscript and the suggestions for improvement.

With this new submission, we believe that we successfully implemented all the proposed changes and hope to meet the requirements to your satisfaction.

Below, we inserted our thoughts and comments into the body of the review letter:

“This paper addresses a topic of growing interest, which is volatility of cryptoasset prices (specifically bitcoin in this paper). While the paper is topical and has potential merit, it possesses too many problems in the form of inaccuracies, omissions, imprecision, and unsubstantiated conclusions to be suitable for publication in its present state. I am very concerned about some of the conclusions in Section III, where the authors attempt to link certain events to changes in bitcoin’s price and fall victim to the very common problem (outside of academia) of mistaking correlation is causation. “

Many previous works \citep{kindleberger,sornette2003,KaizoSor10,BrunnermeierOehmke2013,Xiong2013} have shown that bubbles usually grow out of a rational reaction to a change of economic conditions, or to novel opportunities, new technologies, and so on. Then, the price dynamics amplify beyond what seems justifiable through positive feedback processes. In this spirit, here our purpose is to document what could have been the novel pieces of information that may have nucleated the bubbles. We are well aware that correlation is not causation and our discussion here is more qualitative, with the goal of offering a partial account of the atmosphere in which the bubbles studied quantitatively in the next section developed. This section is thus more descriptive and sets up the backdrop against which to interpret the quantitative findings of section 4.

“For example, the authors state that “The history of Bitcoin at its early stage was highly influenced by fiscal and monetary measures undertaken during the Eurozone crisis” without providing robust empirical evidence to support this view.”

We changed the corresponding sentence to clarify that further below, in fact we do elaborate on the impacts of the Eurozone Crisis and its components on the price of the cryptoasset. Further below, we give again a definition of the Eurozone crisis, in order to show that we are backing the statement. Then, for instance on p.10, as you also point out in the next sentence below, we directly state some of the measures undertaken during the Cypriot Crisis, which in our understanding is part of the Eurozone Crisis. However, we recognise that we do not “provide robust empirical evidence”. Again, this section is not the core of the paper and is offered to present the context in which each of the diagnosed bubbles developed. The description of the context cannot be rigorous but is nevertheless useful to provide the “scenery” of the bubble actions.

“The authors specifically reference the Cyprus deposit haircut crisis as explaining an increase in bitcoin’s price (p. 10 “Cypriot Crises that actually initiated the bubble”).“

We toned down the corresponding statement on p.11, in order to show that the Eurozone and Cypriot Crisis may just have been some of the likely (but not the sole) drivers of the nucleation of this first bubble.

“However, the authors again produce no empirical evidence that buyers of bitcoin were acting on the Cyprus news, only that increased transaction activity and an increase in price coincided with events in Cyprus. “

We recognise that we do not “provide robust empirical evidence”. To our knowledge, it does not seem possible to provide the type of rigorous evidence that the referee is requesting. However, repeating, our goal is to offer a picture of the likely factors that may have been influencing the developments of the diagnosed bubbles.

“In contrast, others have suggested that large buying unrelated to Cyprus in

March-April 2013 by a particular Silicon Valley-based investor is what drove the price increase at this time. “

We thank you for this information and added the following in the text: “It is therefore interesting to observe that the nucleation of the first Bitcoin long bubble occurred at the exact time when the Greece and Cypriot indices reached their troughs. However, we should mention that there have also been rumours that the price increase may have been driven in March-April 2013 by some particular Silicon Valley-based investor. It is even possible that both factors are connected, the savvy investor recognising the implications of the Cyprus-Greek crisis and betting on it by bidding on bitcoin.”

We would like to emphasize again that the point of this subsection is not to demonstrate causality rigorously but to paint the atmosphere of speculation that characterises this period.

“There is also no mention of market manipulation, only changes in fundamental demand, throughout this section as a possible explanation of the price action.”

We now cite two papers suggesting that “Market manipulation, whose presence has been revealed more recently \citep{Gandal_manipul2018,Griffin_manipul2018}, may also have played an important role, for this bubble as well as in the later bubble episodes.”

“I find some of the descriptive terminology employed in the paper confusing. For example, a period of relatively low volatility / flat prices (2015) is referred to as a “normal price growth” phase.”

Thank you for pointing out the misunderstanding that this term could have caused. We have removed this terminology.

“I also question the usefulness of using the term ‘bubble’ throughout the paper given the debate over what constitutes a bubble.”

you are right that defining what is a bubble is a controversial and still not fully settled problem with absence of community consensus. We actually review a part of the literature on this problem. But, in our manuscript, we bypass this problem by making it very clear that we use a specific definition based on the existence of a transient super-exponential regime characterised by the LPPLS pattern. We make this even clearer in p. 4.

“For many ‘bubbles’ or ‘manias’, such as tulips or blatant frauds, we see prices collapse and never rebound.”

It is correct that some bubbles never recover, but we respectfully disagree that this is for “many bubbles or manias”. The historical evidence is clear that stock markets are upward drifting and most stock market bubbles have been followed by a crash and then a recovery period. Anyway, this is a point secondary to our study and do not want to develop it further.

“To my knowledge no other instrument has shown the type of price action as bitcoin. Perhaps the term ‘correction’ would be more useful, appropriate, and less controversial.”

We use the term “drawdown”, which has a sound technical operational meaning. We also use the term crash as it is customary to refer to large corrections as found here.

“In terms of methods, the authors complain of “unpleasant arbitrariness” in the various definitions of bubbles, but then proceed to provide what appears to be arbitrary hurdles for what is a “large peak” and “intermediate peak”, respectively.”

At the end of section II-A, we write “Note that these sets of peak times are robust with respect to significant ($\pm 20\%$) changes of the thresholds $\$0.95\$$ and $\$0.65\$$.” Thus, our approach is not arbitrary.

“I would also be interested in learning how the particular method employed here for dating price regime changes compares with other methods for dating structural breaks, such as Bai and Perron, Banerjee et al, Zeileis et al; no mention of these alternative methods is made in the paper.”

Thank you for this suggestion. Our algorithm for the detection of peak times is just a slight extension of the Epsilon Drawdown Method developed by \cite{JohSoroutlier98,SornetteJohansenoutlier01} and further used in \citep{JohSorBrus10,FILIMONOV201527}. The algorithm is specifically designed to deal with price time series (as opposed to return time series), that is, to deal with their non-stationarity and to detect transient runs of dependence. The performance of the algorithm has thus been documented. In contrast, the methods mentioned by the referee has not well suited for this purpose, as they should apply to stationary time series.

“There are a number of inaccuracies that suggest the authors have an incomplete understanding of bitcoin and its history. For example, in the opening paragraph the authors state “Given the turbulent market history that Bitcoin has undergone since its inception in 2008”, when in fact bitcoins did not have a regularly quoted price or ‘market history’ (meaning they did not trade actively on an exchange) until 2010.”

We apologise for the loose phrasing that led to the misinterpretation mentioned by the referee. We have now removed this confusion and rephrased.

“Later, it is stated inaccurately that “Chinese exchanges such as Huobi and OKCoin moving their business abroad”, when both firms are still quite active (e.g., hiring additional staff) inside China since the “ban”.”

We corrected the statement.

“Some claims relating to China, in addition to being inaccurate, lack novelty.”

To repeat, the purpose of this section is to provide a self-contained synthesis of what we consider the main atmosphere and environment prevailing at the time of the bubble regimes that we diagnose. Our intention is not about novelty but about setting the stage to interpret better the quantitative analysis presented in the following section.

“The source of the price data is not provided and needs to be provided, along with other information (e.g., frequency).”

We inserted a sentence for this. The source of the data is Thomson Reuters Datastream Bitstamp quoted daily data from August 2011.

“Better referencing is also required (e.g., p. 3 which pundits have compared bitcoin to tulips and in what source?) There is also a disconcerting level of imprecision for a scientific paper (e.g., p. 2 “loss from the peak of about two thirds” – the actual figure is 70%).”

We fixed the two inaccurate statements in p.2 and added a source on p.3 which explicitly compares bitcoin and tulip bubble price series in a plot.

“A suggestion: some additional data in the tables would also be useful (e.g., see below example for other useful data). Also, some of the below data does not agree with the paper data, but I cannot validate as I do not know what price data source the author is using.”

We have added more information in Table 1, namely two columns giving additional absolute price information on all qualified bubbles. Indeed, especially regarding the immense growth of Bitcoin's price over the past years, your suggestion to add some absolute price data for better orientation, was very helpful.

Appendix C

Review of Bitcoin's multiscale bubble history from January 2012 to February 2018

Dear Reviewer and Editors,

thank you for taking the time to review our manuscript for the second time. The suggestions for improvement are greatly appreciated. With this ultimate submission, we believe that we successfully implemented all the proposed changes to our best possibilities and hope to meet the requirements to your satisfaction. Below, we inserted our thoughts and comments into the body of the review letter:

Overall summary

This paper describes the use of the classical log-periodic power law model to analyse speculative bubbles in Bitcoin markets – itself an emerging topic of independent research interest in finance and economics. Whilst I have some concerns that more advanced methods are ultimately possible, e.g. based on extensions of the model given in Appendix B, I think it would be important to see this group publish their perspective of such an important topic in a good journal such as this.

Comments on previous revision

I felt that some of the previous reviewer's comments were a bit too harsh e.g. the application of speculative bubble models to Bitcoin and cryptocurrency data is now well-established. As is the application of econophysics models such as this. See for instance Cheah and Fry (2015), Fry and Cheah (2016) and Fry (2018). Note that these are papers published in mainstream economics and finance journals – here the publications in the economics journal are particularly noteworthy. Whilst I am easy-going about whether the authors choose to cite these in a revision the authors might find Fry and Cheah (2016) useful as a source of references both on previous econophysics models applied to Bitcoin and the early economics and finance literature on Bitcoin.

Thank you for pointing out these articles which make a good addition to our list of references. The corresponding references are cited in the paper, now.

A wholesale check of the article references should be performed e.g. the reference by Fantazzini [10] is already published. Similarly, the article by Glazer et al. has been published in a journal.

We have rechecked and correctly referenced all sources to the best of our knowledge.

Required changes

Some more details of the Lagrange regularisation approach are needed. This seems to me to be a penalised regression approach with a penalty applied to penalise against choosing a time window that is too short?

Not exactly. The difference in the approach is that the penalty is applied after obtaining the regression results and not during the regression by penalizing the cost function. The Appendix Section D describing the technique was complemented with some more details that should clarify the procedure.

Section 2.1.

More explanation is required as to the choice of ϵ_0 and w . Why were these numbers chosen? More explanation is also needed as regards how the short-bubble/long-bubble typology depends on the stated value of f_t . It is not clear where this comes from. In principle, I like this distinction between short-bubbles and long-bubbles but I think this needs to be placed on a firmer foundation.

The text may not have been completely clear about the procedure which is why we refer the reader to the corresponding appendix explaining in detail the mathematical backgrounds of the epsilon method. The reason that we put this and other explanations like the lagrange regularisation or the lplls estimation to the appendix is that we would like to keep the flow of the text. We however have adapted the text paragraph summarizing the epsilon metric now, in order to make it more understandable also without reading the appendix. This should also clarify how the values of epsilon and w are robustly 'averaged out' from the procedure and that the

outcome of the complete procedure with respect to the choice of the threshold value f_t is actually quite robust.

I like the idea behind Section 3. This might need a bit better positioning in terms of

- (a) Justified as a historical study of Bitcoin the archetypal cryptocurrency and of independent financial and economic interest in its own right
- (b) Similar analyses have already been undertaken before (see e.g. Fry and Cheah, 2016)

We have revisited the section and added a few more comments in different places. We would like to clarify that the overall intention of this section mainly is to justify the results of the bubble start and peak analysis that was technically determined in Section 2 from an event point of view. We particularly searched for events that may have fueled bubble formation and growth during the identified bubble periods. As mentioned in the document, it is by no means the goal to cover the complete history of Bitcoin here, but to list a variety of likely explanations for the intense bubble growth periodically observed on the cryptocurrency. This results in a long section of explanations, however the essential part of the paper then follows in the predictive analysis using the LPPLS model.

There are necessarily some subjective elements here as identified by the previous reviewer but I don't think that this is necessarily a problem. Need to provide a reference for the comment about a Silicon Valley based investor.

Done.

I think the orientation of Section 4 is wrong and this section should be replaced with something more in keeping with the overall aims of the paper. If this paper is for the most part a historical analysis of Bitcoin then I think this issue of real-time analysis is slightly tangential and out of place.

It is important to us to mention that it is, as stated, not the sole goal of the paper to give a complete historical overview over the evolution of Bitcoin. Rather, we use section three, in order to justify our technically determined bubble results, in order to show that indeed the results based on LPPLS analysis agree with real world events. We consider the real time analysis an essential and valuable core piece of the paper, as we can show here that indeed with the LPPLS model we are able to predict regime changes in Bitcoin.

What I think would be more useful in its place is to include an additional robustness check of the long and short bubbles identified in Section 3 using a more standard econometric methodology. Possibly something as simple as a standard event study. I was a little concerned about how the referee comment about using alternative methods was seemingly casually dismissed in the original rebuttal. The method of Bai and Perron is implemented in an R package, *Strucchange* I think, and might be used to test for a change in the mean or the variance of the log-returns. However, I suspect a better approach here would be to use a simple event study methodology applied to the start-date of each of the identified short and long bubbles.

We have added an additional section in the appendix, where we perform a change point detection on the returns of Bitcoin during the analyzed time period, similar to the original suggestion by the reviewer to employ the method of Bai and Perron. We find that the main regimes identified with the change point analysis agree with our results. We did not include an event study, as we consider it a difficult practice to try to assign bitcoin returns to specific events. This is why we on the one hand throughout the paper emphasize again and again that the listed nucleating factors are just some of the possible influences that led to bubble formation, and on the other hand, we emphasize that the bubble start times that we find are the dates that initiate a period during which multiple events that promote bubble growth may occur.

The conclusions section should be shortened and better focussed to the contribution of the paper – namely a retrospective analysis and typology of historical bitcoin bubbles by somebody whose work I've always admired.

We have shortened the conclusion section.

Do check the references as I think there are probably multiple mistakes here. (I realise this may be harder to get right than in other papers due to the abundance of electronic-only references and the fast-paced nature of the subject).

As stated above, we have rechecked all references. In order to avoid confusion about the online sources, we have created a dryad repository for storage of all relevant online articles in pdf format. The repository can be openly accessed by anyone trying to track the sources.

Sincerely,

Jan-Christian Gerlach, Dr. Guilherme Demos and Prof. Didier Sornette.

Appendix D

Response to final review, May 2019

Dear Reviewer and Editors,

Thank you for the final review of our paper. We are glad to hear that the manuscript was accepted for publication at RSOS. In this final iteration of our paper, we have introduced minor additional changes, as requested by the reviewer and editors. Please see our direct reply to the review letter below inline.

Review of Bitcoin's multiscale bubble history from January 2012 to February 2018

Overall comments

I would be happy to recommend publication of a suitably revised manuscript. I would also like to acknowledge the efforts of the author team in revising the previous version. However, there are a small number of minor changes I would like to see first. The main thing here is that I think a couple of minor changes in writing style would be more in-keeping with a finance paper.

Thank you.

Required changes

Section 2.1

This now reads much more clearly. I think it would be worth just adding a sentence on the financial interpretation of these parameters. For example, ϵ_0 accounts for anything up to a 5 standard deviation event and for a regular financial market that trades 5 days a week w corresponds to intervals of between 2 weeks to 12 weeks.

We added some explanation to the paragraph in the text body. We point out, that for detailed specifications of the described techniques, the reader should always refer to the Appendices referenced in the text, where everything is described in full detail.

Appendix E

I like the Appendix E. However, I think this would be better in the main manuscript e.g. Section 4.3 Additional robustness check. This would give your article more authority in terms of a finance article. It also seems a shame to have this work hiding at the end of an Appendix.

Thank you for pointing out the value of this part which in fact is better placed in the text body. We inserted the appendix E (with a little adaptation) as the new subsection 2.4 and deleted the corresponding Appendix. We think that the content is best suited directly following the determined bubbles, because it relates to and confirms the presented results.

References

Could you check references 10, 20 and 41 and possibly the remainder. One of the things to point out here is that SSRN is an online repository rather than an electronic journal and these articles may be more properly regarded as preprints. Do check though that these articles have not been formally published in a journal.

We have checked all references again. To the best of our knowledge, all references are up to date and we did not find any newer versions or citations.

Conclusions section

In order to bolster the finance contribution of the paper I think it would be worthwhile to clarify in the first paragraph of the conclusions section:

- From an academic perspective the study of Bitcoin as the archetypal cryptocurrency is important from an economic and a finance perspective.
- Approach also adds to econophysics analyses of the area that have been conducted before

The conclusion section now comprises a new first paragraph stating the reasons why we believe it is worthwhile to study Bitcoin as a representative example. This addresses your first point. Furthermore, another paragraph has been added further below, emphasizing the relevance of the quantitative LPPL methods applied in this paper, thereby addressing your second point.

Possible suggestion

The conclusions section is now much more clearly written. As regards the final paragraph you could consider reading Dowd (2014) where one of the concerns raised is that Bitcoin may ultimately prove vulnerable to competitors. Another issue that may be worth considering is the ultimate vulnerability of these currencies to national regulation – e.g. the new Swiss e-Franc, new cryptocurrencies backed by large investment banks intended to supersede Ripple etc.

Dowd, K. (2014). New private monies – a bit-part player? Institute for Economic Affairs, London.

Thank you for the additional source. We retrieved the book and read it in details. We decided not to add it as a reference because, while interesting and important, it is not relevant to our paper: please note again, that although the social bubble drivers section is a contribution to our paper, our main focus is on the quantitative techniques that we use to identify bubbles ex-ante and ex-post and to predict crashes. Additionally discussing the legal and regulation aspects behind Bitcoin in the conclusions section would open up another topic, deviating the attention away from the main points of our study.

Concerning the point about vulnerability to competitors, we already pointed out in the social drivers section that the market capitalization share of Bitcoin has dramatically decreased in 2017/18, because of new money flowing into competing ICO-released CCs. We believe that this point has therefore been addressed.

With kind regards,

Gerlach, Demos and Sornette